# Cycle-of-Science: Reliable Reasoning through Counterfactual Verification for Agent Decision Making

**Ruojie Zhang**[1]  **Wencheng Zhu**[2]  **Peiyuan Jiang**[1]  **Dayong Zhu**[1]

## Abstract

Large Language Models have significantly advanced autonomous agents through their sophisticated perception and execution capabilities. Despite effective, agents still struggle with robust decision-making due to passive learning from similar experiences that often confound correlation with causality. Inspired by the Scientific Method, we propose a Cycle-of-Science framework that autonomously explores potential causal pathways through an iterative loop of *Hypothesis, Experiment, and Validation*, enabling agents to identify truly effective causal dependencies. To be specific, we first leverage causal knowledge to guide the initial hypotheses generation. These hypotheses are then analyzed through experiments using counterfactual samples. Afterward, we perform causal analysis to quantify effects of interventions, deriving well-validated hypotheses for next agent steps. To train our policy, we further introduce a two-stage pipeline that integrates supervised fine-tuning with Counterfactual Preference Optimization, which constructs preference signals from intervention outcomes to reinforce validated reasoning chains. Experiments on benchmarks demonstrate that our method achieves superior performance over state-of-the-art approaches.

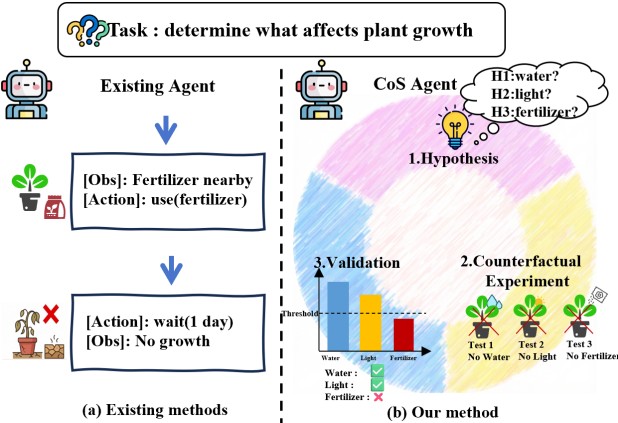

*Figure 1.* Comparison of existing methods and our framework. Existing agents rely on passive learning from statistical patterns, leading to spurious correlations. In contrast, our framework autonomously explores causal pathways through hypothesis generation, counterfactual experimentation, and intervention validation.

## 1. Introduction

The integration of LLMs as the reasoning and decision module of interactive agents has achieved substantial progress across several decision-making applications (Li et al., 2024; Liu et al., 2025; Wang et al., 2024a; Romero et al., 2023),

including unmanned aerial vehicles (Zeng et al., 2016; Hu et al., 2023), autonomous driving (Wang et al., 2023; Mao et al., 2023), and robotic manipulation (Ahn et al., 2022; Brohan et al., 2023).

The main objective is to transform LLMs from text processors into autonomous decision-making agents capable of environmental perception and strategic action execution. To achieve this goal, many LLM-based agent approaches have been proposed broadly categorized into three categories, including i) *Prompt-based methods* (Wei et al., 2022; Yao et al., 2023b;a) elicit decision-making through thought-action prompting sequences with frozen LLMs. ii) *Experience-driven methods* (Shinn et al., 2024; Zhao et al., 2024; Madaan et al., 2023) refine agent policies through iterative feedback and summaries of past interaction trials. iii) *Hierarchical planning methods* (Ahn et al., 2022; Song et al., 2023) enable structured decision-making by decomposing long-horizon tasks into manageable sub-goals.

Existing approaches mainly mirror a pattern-matching process optimized for the imitation of observed trajectories (Pomerleau, 1991; Ross et al., 2011) rather than the acquisition of the underlying causal logic. These paradigms lack

---

[1] University of Electronic Science and Technology of China, Chengdu, China  [2]School of Artificial Intelligence, Tianjin University, Tianjin, China. Correspondence to: Wencheng Zhu <wenchengzhu@tju.edu.cn>, Dayong Zhu <cnzdy@uestc.edu.cn>.

*Proceedings of the 43$^{rd}$ International Conference on Machine Learning*, Seoul, South Korea. PMLR 306, 2026. Copyright 2026 by the author(s).

explicit mechanisms to disentangle causality from statistical correlation (Schölkopf et al., 2021; Peters et al., 2017), resulting in decision-making fragility when encountering with complex confounding factors (Pearl, 2009; Zhang et al., 2021). Consider the ScienceWorld scientific environment (Wang et al., 2022) in Figure 1a, when asked to "*determine what affects plant growth*," correlation-based agents often execute `use(fertilizer)` based on the observation that fertilizer frequently appears near growing plants in training trajectories. This heuristic overlooks essential preconditions that fertilizer is conditional on the presence of adequate water, light exposure, and soil quality. While spatial proximity often correlates with goal achievement, it is not a sufficient causal determinant of the desired outcome. Furthermore, without identifying true causal factors, agents are forced into trial-and-errors with high complexity (Kakade, 2003; Lattimore & Szepesvári, 2020). Consequently, an ideal decision-making requires more than just passive observation or reinforcement, it necessitates an active mechanism to verify the validity of reasoning before execution (Han et al., 2024; Sun et al., 2025). In human cognition, this is achieved through scientific reasoning that treats internal beliefs as testable conjectures (Gopnik et al., 2004; Sloman, 2005). With a "scientist's mindset," an agent can move from asking "What usually happens next?" to "Why does this action lead to success?".

Inspired by this, we propose a Cycle-of-Science framework that structures agent decision making as iterative causal reasoning loops. When facing the ScienceWorld plant growth task, our method instead employs a rigorous cognitive process shown in Figure 1b. Specifically, our framework includes three stages, including *Hypothesis Generation*, where agents enumerate potential causal factors, such as verifying whether light, water, or fertilizer independently facilitates growth, and produce hypotheses; *Counterfactual Experimentation*, involving designing controlled interventions to isolate individual variables, e.g., providing light without fertilizer; and *Intervention Validation*, where agents quantify the causal impact of each intervention to distinguish effective actions from spurious correlations. In summary, contributions of our method are summarized as three-aspects:

- We propose an intervention-based Cycle-of-Science framework that structures agent decision making into a Hypothesis-Experiment-Validation loop.

- We develop a preference training pipeline that combines supervised fine-tuning with Counterfactual Preference Optimization, using intervention outcome comparisons instead of human annotations.

- We provide a comprehensive empirical study on ScienceWorld, ALFWorld, and Crafter, and experimental results demonstrate superior performance over state of-the-art approaches.

## 2. Related Work

### 2.1. LLMs as Decision-Making Agents

LLMs have demonstrated strong decision-making capabilities for autonomous agents across reasoning, robotics, and interactive environments (Li et al., 2024; Ahn et al., 2022; Wang et al., 2022). Current approaches can be broadly classified into three categories: structured prompting methods (Wei et al., 2022; Yao et al., 2023b;a) decompose tasks via chain-of-thought or tree-search; experience-based methods (Shinn et al., 2024; Zhao et al., 2024; Madaan et al., 2023) leverage reflective feedback or trajectory retrieval to refine policies; hierarchical planning methods (Ahn et al., 2022; Song et al., 2023; Sun et al., 2023) coordinate multi-level goal execution. While effective, these methods rely on passive observation and learn patterns from historical demonstrations without a grasp of the underlying action-effect logic. This often confounds correlation with causality. When agents encounter new scenarios with different causal structures, learned patterns fail to generalize. We shift the paradigm from passive learning to active causal exploration through a loop of hypothesis generation, counterfactual experimentation, and validation.

### 2.2. Causal Reasoning in LLMs

Causal reasoning is important for LLMs to predict intervention outcomes beyond recognizing statistical correlation (Pearl, 2009; Schölkopf et al., 2021). Early work (Kıcıman et al., 2024; Jin et al., 2023b) explores LLMs' ability to encode causal knowledge from pre-training. Jin et al. (2023a) formalize causal inference across Pearl's causal ladder, finding that LLMs struggle with complex counterfactual reasoning on basic symbolic tasks. Recent efforts have shifted toward constructing causal graphs using LLMs. Long et al. (2023) and Ban et al. (2023) leverage LLMs to extract causal relationships from text by retrieving relevant passages and identifying factor similarities. However, these methods are limited without systematic causal discovery strategies, and hallucination remains due to lack of validation (Wang et al., 2024b). Chen et al. (2025) propose a causal-aware framework that learns causal structures from interaction histories, but relies on passive observation and outcome feedback rather than quantified causal effect estimation through controlled experimentation (Pearl, 2009). In contrast, our method emphasizes decision-time intervention testing for action selection. A related line of work investigates LLMs for scientific hypothesis generation and experimental design (Cohrs et al., 2025; Li et al., 2025; Manning et al., 2024), but focuses on static knowledge extraction rather than interactive decision-making with intervention validation. These limitations motivate us to harness the reasoning capabilities of LLMs through an iterative cycle of hypothesis generation, counterfactual experimentation, and quantitative validation.

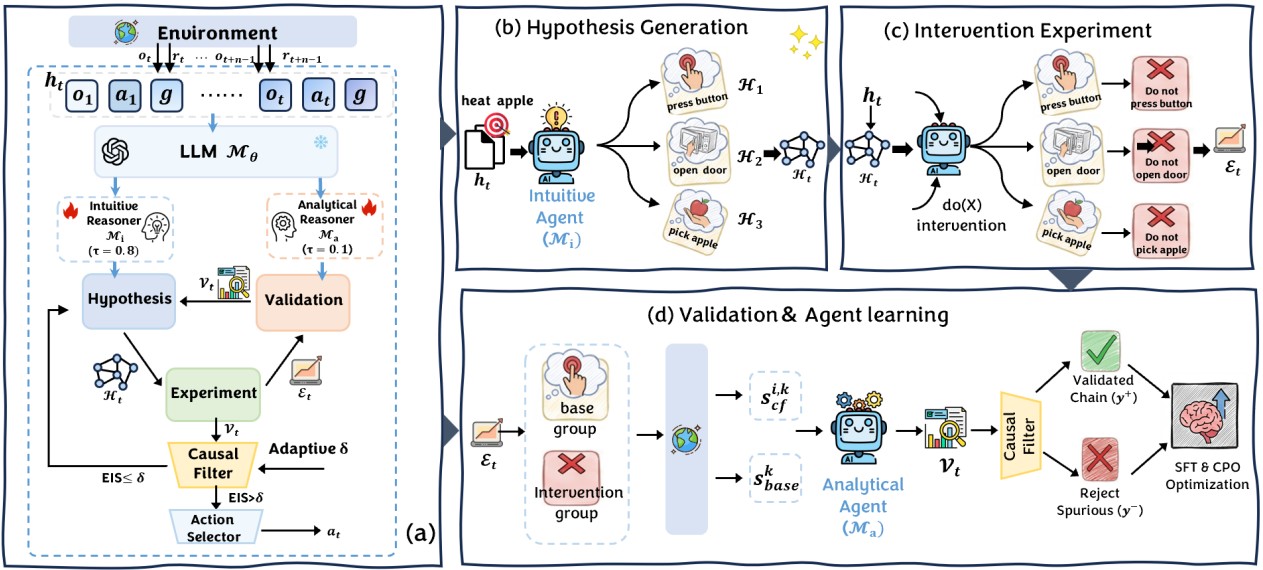

*Figure 2.* **Overview of the Cycle-of-Science framework. (a) Framework Pipeline.** The agent executes a Hypothesis-Experiment-Validation loop with dual reasoners including Intuitive $\mathcal{M}_i$ and Analytical $\mathcal{M}_a$, and causal filtering to select actions. **(b) Hypothesis Generation**: The Intuitive Agent generates multiple causal hypotheses $\{\mathcal{H}_1, \mathcal{H}_2, \mathcal{H}_3, ...\}$ based on historical trajectory $h_t$. **(c) Intervention Experiment**: For each hypothesis $\mathcal{H}_i$, the agent performs counterfactual interventions $\text{do}(X)$ and observes outcomes $\mathcal{E}_t$. **(d) Validation & Agent Learning**: The Analytical Agent computes EIS and filters hypotheses by threshold $\delta$, producing validated causal chains $\mathbf{y}^+$ for agent optimization while rejecting spurious correlations $\mathbf{y}^-$.

## 3. Method

### 3.1. Preliminary

We model agent interactions with environment as a Markov Decision Process $\langle \mathcal{S}, \mathcal{A}, \mathcal{P}, \mathcal{R}, \gamma \rangle$, where $\mathcal{S}$ and $\mathcal{A}$ represent the state and action spaces. $\mathcal{P} : \mathcal{S} \times \mathcal{A} \times \mathcal{S} \rightarrow [0, 1]$ denotes the transition probability, $\mathcal{R} : \mathcal{S} \times \mathcal{A} \rightarrow \mathbb{R}$ is the reward function, and $\gamma \in (0, 1]$ is the discount factor. In this work, we consider an LLM-based policy $\pi_\theta : \mathcal{S} \times \mathcal{A} \rightarrow [0, 1]$ parameterized by pretrained language model weights $\theta$, where states $s \in \mathcal{S}$ are represented as natural language observations and actions $a \in \mathcal{A}$ are textual commands. The objective is to maximize the expected return $J(\pi_\theta) = \mathbb{E}_{\pi_\theta}[\sum_t \gamma^t \mathcal{R}(s_t, a_t)]$ via iterative causal reasoning.

### 3.2. The Cycle-of-Science Framework

Our framework includes three components, comprising a Hypothesis Generator (CHG), an Intervention Planner (IP), and an Effect Validator (EV). An overview of the full pipeline is shown in Figure 2.

**Causal Hypothesis Generator.** The Hypothesis Generator employs an Intuitive Reasoner $\mathcal{M}_i$ derived from the base language model $\mathcal{M}_\theta$ with high-temperature sampling ($\tau = 0.8$) to promote divergent exploratory reasoning. At timestep $t$, CHG takes the historical context $h_t = [o_1, a_1, \ldots, o_t, g]$ as input encoding past observations, actions, and the objective goal, and generates a set of can-

didate hypotheses $\mathcal{H}_t = \{(C_i, E_i)\}_{i=1}^N$, where each tuple specifies a potential action $C_i$ and its high-level intent $E_i$ toward achieving the goal. The high-temperature decoding enables CHG to sample diverse hypotheses from the current policy $\pi_\theta$, ensuring broad exploration of the action space. For example, given the objective to heat food in a microwave, CHG may hypothesize actions such as *"press the start button"* or *"close the door"*. This divergent reasoning phase provides robust action candidates for subsequent experimental verification.

**Intervention Planner.** The Intervention Planner converts each candidate hypothesis into a contrastive intervention test for online decision making. For each hypothesis $(C_i, E_i) \in \mathcal{H}_t$, we create two rollout arms from the same decision state: (i) an intervention arm that executes the candidate action $C_i$, and (ii) a baseline arm that does not apply this target action at that step. This setup directly operationalizes our decision-time comparison as *action applied* versus *action not applied* under matched initial conditions. To avoid ambiguity, this procedure should be interpreted as a controlled environment intervention for hypothesis testing, rather than a formal negation-based counterfactual in a structural causal model. Accordingly, IP outputs a set of test protocols $\mathcal{E}_t = \{(C_i, E_i)\}_{i=1}^N$, which are passed to the Effect Validator for quantitative scoring and filtering.

**Effect Validator.** The Effect Validator utilizes an Analytical Reasoner $\mathcal{M}_a$ instantiated from the base language model

$\mathcal{M}_\theta$ using low-temperature sampling ($\tau = 0.1$) for rigorous evaluation. The EV evaluates intervention designs by computing intervention effect scores that quantify their impact within a specific task context. For each hypothesis, EV performs controlled experiments by comparing the outcomes of an intervention group against a baseline group. Specifically, to ensure ceteris paribus, we preserve the current state $s_t$ and perform multiple rollouts. The intervention group executes the proposed action from this state (denoted as $s_{\text{int}}^{i,(k)}$), while the baseline group follows a no-intervention policy (denoted as $s_{\text{base}}^{(k)}$). By resetting the environment to the same initial state before each trial, this protocol guarantees that both groups share matched initial conditions for a fair comparison. EV then quantifies intervention impact by comparing expected outcomes under intervention against baseline:

$$\text{EIS}_i = \frac{1}{K} \sum_{k=1}^{K} f_i\left(s_{\text{int}}^{i,(k)}\right) - \frac{1}{K} \sum_{k=1}^{K} f_i\left(s_{\text{base}}^{(k)}\right). \quad (1)$$

Here, $\text{EIS}_i$ denotes the empirical intervention score for hypothesis $C_i$. The evaluation function $f_i(s)$ utilizes $\mathcal{M}_a$ to score goal achievement based on the expected effect description $E_i$. $s_{\text{int}}^{i,(k)}$ represents the state observed after executing the $i$-th intervention from the initial state $s_t$, and $s_{\text{base}}^{(k)}$ represents the state in the baseline group with no intervention from the same initial state $s_t$. Importantly, EIS is used as a relative ranking signal for hypothesis filtering within the current decision state, rather than a calibrated probability or a globally identified causal quantity. Therefore, the validator focuses on local action-effect verification for decision making. To balance statistical reliability with computational efficiency, we perform $K = 3$ independent rollouts for each condition.

### 3.3. Intervention Validation Criterion

We establish a intervention validation criterion to filter hypotheses and guide action selection. Given a set of candidate hypotheses $\mathcal{H}_t = \{(C_i, E_i)\}_{i=1}^{N}$ and their corresponding causal effects $\mathcal{V}_t = \{\text{EIS}_i\}_{i=1}^{N}$ measured by the EV, each hypothesis is classified as follows:

- **Valid Hypothesis.** When $\text{EIS}_i > \delta_t$, the hypothesis have causal impact and is retained for decision-making;

- **Invalid Hypothesis.** When $\text{EIS}_i \leq \delta_t$, the hypothesis relies on spurious correlations and is rejected.

However, a static threshold fails to account for the variability of decision contexts because the median EIS for successful actions empirically ranges from 0.23 on ALFWorld to 0.71 on ScienceWorld. To handle this, we introduce an adaptive threshold modulation strategy, drawing inspiration

from uncertainty-driven exploration principles (Pathak et al., 2019; Haarnoja et al., 2018).We quantify decision ambiguity through the action entropy of the policy as

$$H(a_t|h_t) = -\sum_{a \in \mathcal{A}} \pi_\theta(a|h_t) \log \pi_\theta(a|h_t), \quad (2)$$

where $\pi_\theta(a|h_t)$ is the probability of selecting action $a$ given the historical context $h_t$ under policy $\pi_\theta$. Accordingly, the baseline threshold $\delta_{\text{base}}$ is dynamically adjusted as

$$\delta_t = \delta_{\text{base}} \cdot \left(1 - \lambda \cdot \frac{H(a_t|h_t)}{H_{\max}}\right), \quad (3)$$

where $\delta_{\text{base}} = 0.5$ is the baseline threshold and $\lambda = 0.3$ is the modulation coefficient. We normalize the entropy using $H_{\max} = \log|\mathcal{A}|$ and $|\mathcal{A}|$ denotes the size of the action space. In uncertain states with high $H(a_t|h_t)$, the threshold is lowered to encourage the exploration of a broader range of causal hypotheses; in confident states with low $H(a_t|h_t)$, the mechanism maintains a strict filtering criterion.

### 3.4. Dataset Preparation and Training

**Data Construction.** We collect base trajectories from human demonstrations across 100 tasks. For each decision step, we augment the trajectory by executing the CHG-IP-EV cycle: generating hypotheses, designing interventions, and quantifying causal effects through environment rollouts. To ensure high-quality training data, we retain only those trajectories where validated hypotheses ($\text{EIS} > \delta_t$) directly align with successful task completion (reward $> 0.8$). This process yields a refined dataset, denoted as

$$\mathcal{D}_c = \bigcup_{m=1}^{M} \bigcup_{t=1}^{T_m} \{(h_t, \mathcal{H}_t, \mathcal{E}_t, \mathcal{V}_t, a_t, o_{t+1})\}, \quad (4)$$

where each tuple contains the historical context, candidate hypotheses, measured causal effects, action, and next observation. This procedure initially yields 5K seed trajectories. To further scale the data, we expand the dataset to 50K trajectories via self-play, where the partially trained model generates new trajectories following the same annotation and verification protocol.

**Two-Stage Training.** We employ a two-stage training pipeline on $\mathcal{D}_c$. In the first phase, we perform supervised fine-tuning on the base model $\mathcal{M}_\theta$ to generate reasoning chains $\mathbf{c}_t = (\mathcal{H}_t, \mathcal{E}_t, \mathcal{V}_t)$. The objective is formulated as:

$$\mathcal{L}_{\text{SFT}} = -\mathbb{E}_{(h_t, \mathbf{c}_t, a_t) \sim \mathcal{D}_c} \log \pi_\theta(\mathbf{c}_t, a_t|h_t). \quad (5)$$

This stage produces a reference model $\pi_r$ that serves as the initialization for the subsequent optimization. We then construct a causal preference dataset $\mathcal{D}_{\text{pref}}$ by sampling multiple chains from $\pi_{\text{ref}}$ for each state $h_t$. These chains are ranked

according to their measured causal effects: those with EIS $> \delta_t$ are labeled as positive ($y^+$) while those with EIS $\leq \delta_t$ are negative ($y^-$). Finally, we introduce Counterfactual Preference Optimization (CPO) to maximize the likelihood margin between causally valid and invalid reasoning paths. The optimization objective is defined as

$$\mathcal{L}_{\text{CPO}} = -\mathbb{E}_{\mathcal{D}_{\text{pref}}} \left[ \log \sigma \left( \beta \log \frac{\pi_\theta(y^+|h)}{\pi_{\text{ref}}(y^+|h)} - \beta \log \frac{\pi_\theta(y^-|h)}{\pi_{\text{ref}}(y^-|h)} \right) \right],$$
(6)

where $\beta = 0.1$ controls the deviation from the reference model $\pi_{\text{ref}}$ and $\sigma$ denotes the sigmoid function. To stabilize training and mitigate overfitting toward shorter outputs, we incorporate a length-normalized regularization term:

$$\mathcal{L}_{\text{NLL}} = -\mathbb{E}_{(h,y^+)\sim\mathcal{D}_{\text{pref}}} \frac{\log \pi_\theta(y^+|h)}{|y^+|},$$
(7)

where $|y^+|$ represents the token length of the preferred output. The final training objective combines both terms as $\mathcal{L} = \mathcal{L}_{\text{CPO}} + \alpha\mathcal{L}_{\text{NLL}}$ with $\alpha = 0.01$. A primary contribution of our approach lies in the construction of $\mathcal{D}_{\text{pref}}$ using empirically measured causal effects rather than human preferences, enabling the model to learn from environment-validated causal reasoning.

## 4. Experiments

We conduct extensive experiments to evaluate our framework, specifically addressing the following questions:

- **RQ1 (Performance):** How does CoS compare with SOTA benchmarks? We evaluate performance on ScienceWorld, ALFWorld, and Crafter in Sec. 4.2.

- **RQ2 (Ablation):** What are contributions of components? We ablate the scientific reasoning loop and the training strategies in Sec. 4.3.

- **RQ3 (Hyper-parameter):** How do design choices impact effectiveness? We analyze the adaptive threshold and hypothesis diversity in Sec. 4.4.

- **RQ4 (Visualization):** How does CoS reason in practice? We provide case studies comparing CoS with ReAct and KnowSelf on ALFWorld tasks in Sec. 4.6.

### 4.1. Experimental Settings

**Datasets.** We evaluate CoS on three interactive decision-making benchmarks: 1) *ScienceWorld* (Wang et al., 2022) is a text-based scientific simulation with 30 tasks spanning physics, chemistry, and biology. Agents must perform experiments by manipulating variables and observing outcomes to complete scientific objectives. 2) *ALFWorld* (Shridhar et al., 2021) consists of six categories of household manipulation tasks. This environment requires agents to perceive textual

descriptions, navigate environments, and interact with objects to fulfill high-level instructions. 3) *Crafter* (Hafner, 2022) is a 2D survival game ($64 \times 64$ grid, $9 \times 7$ view) with partial observability. The environment contains natural resources (trees, stone, iron), hostile entities (zombies), and craftable items. Agents are required to collect and create items to unlock a hierarchical tree of 22 total achievements.

**Baselines.** We evaluate three open-source backbones: Mistral-7B (Jiang et al., 2023), Gemma-7B (Gemma Team et al., 2024), and Llama-3-8B (Dubey et al., 2024). For fair comparison, we follow standard evaluation protocols with hyperparameters and prompts detailed in the Appendix. We compare against five baselines: *ReAct* (Yao et al., 2023b) interleaves reasoning traces with action execution through prompt-based guidance. *Reflexion* (Shinn et al., 2024) incorporates self-reflective verbal feedback from past failures for decision-making. *ExpeL* (Zhao et al., 2024) retrieves relevant success/failure experiences from past trajectories to guide current tasks. *KnowSelf* (Qiao et al., 2025) aligns internal knowledge with external feedback. *GLIDER* (Hu et al., 2025) employs hierarchical planning by dividing tasks into sub-goals to reduce action space complexity.

### 4.2. Main Results

**Performance on ScienceWorld.** As shown in Table 1, CoS achieves the best performance compared to baselines on both seen and unseen splits across three backbone models (Mistral-7B, Gemma-7B, and Llama-3-8B). The improvements over GLIDER range from +7.37% to +12.69% on seen tasks and +6.44% to +12.91% on unseen tasks. For instance, with Mistral-7B on ScienceWorld, CoS achieves 71.45 on seen tasks and 67.23 on unseen tasks, outperforming GLIDER by +12.69% and +12.91% respectively. The substantial gains on unseen tasks highlight CoS's ability to identify genuine causal mechanisms rather than superficial correlations. This is crucial for reliable agent behavior in novel environments. Beyond accuracy, CoS demonstrates superior step efficiency, completing tasks within 61.8–68.5 steps with a 7.9%–9.7% reduction compared to GLIDER's 68.1–75.9 steps. The hypothesis-driven exploration is more efficient than trial-and-error because agents avoid wasteful interactions by identifying plausible causal drivers before engagement.

**Performance on ALFWorld.** On ALFWorld, CoS demonstrates strong generalization on unseen tasks but modest improvements on seen tasks. For unseen splits, CoS consistently outperforms baselines across all backbones, with improvements ranging from +2.16% to +2.26% over GLIDER. For instance, with Llama-3-8B, CoS achieves 84.76% on unseen tasks compared to GLIDER's 82.50%. However, on seen tasks, the improvements are limited, with CoS achieving competitive but not universally superior performance.

*Table 1.* **Performance Comparison on ScienceWorld and ALFWorld.** Success rate (%) and average steps for completion across different backbones. *Prompt-based methods; †Fine-tuning methods with LoRA. Bold indicates best performance in each column.

| Backbone | Method | ScienceWorld | | | ALFWorld | | |
|---|---|---|---|---|---|---|---|
| | | Seen | Unseen | Steps | Seen | Unseen | Steps |
| Mistral-7B | ReAct* | 20.72 | 17.65 | 127.3 | 58.35 | 54.21 | 34.2 |
| | Reflexion* | 26.34 | 23.18 | 98.5 | **84.24** | 79.86 | 22.1 |
| | ExpeL* | 35.67 | 32.45 | 89.2 | 78.68 | 74.12 | 21.8 |
| | KnowSelf† | 52.18 | 48.93 | 74.6 | 81.41 | 76.84 | 19.2 |
| | GLIDER† | 58.76 | 54.32 | 68.1 | 82.61 | 78.25 | 18.7 |
| | CoS (Ours)† | **71.45** | **67.23** | **61.8** | 83.28 | **80.51** | **17.3** |
| Gemma-7B | ReAct* | 3.58 | 3.21 | 145.7 | 30.65 | 27.18 | 28.9 |
| | Reflexion* | 8.91 | 7.54 | 118.4 | 48.70 | 44.35 | 26.4 |
| | ExpeL* | 18.42 | 16.73 | 105.3 | 40.51 | 36.89 | 25.7 |
| | KnowSelf† | 45.63 | 42.17 | 82.5 | 70.96 | 66.42 | 21.3 |
| | GLIDER† | 52.89 | 49.26 | 75.9 | **79.34** | 72.91 | 20.1 |
| | CoS (Ours)† | **64.82** | **60.47** | **68.5** | 77.25 | **74.18** | **19.2** |
| Llama-3-8B | ReAct* | 24.76 | 22.31 | 132.1 | 40.01 | 36.54 | 27.6 |
| | Reflexion* | 32.45 | 29.87 | 103.7 | 57.14 | 52.68 | 25.2 |
| | ExpeL* | 41.28 | 38.65 | 94.8 | 49.27 | 45.13 | 24.8 |
| | KnowSelf† | 56.83 | 53.19 | 76.4 | 84.53 | 79.17 | 20.4 |
| | GLIDER† | 62.47 | 58.91 | 69.7 | 81.28 | 76.42 | 19.8 |
| | CoS (Ours)† | **69.84** | **65.35** | **64.2** | **84.76** | **80.58** | **18.1** |

*Table 2.* **Scores (mean ± std) on 22 Crafter tasks.** Score represents the percentage of achievements unlocked (out of 22 total). All methods are evaluated over 1M environment steps.

| | Method | Score (%) |
|---|---|---|
| RL-based | Rainbow(@1M) | 19.5 ± 1.8 |
| | DreamerV3(@1M) | 17.3 ± 1.9 |
| | SPRING(@1M) | 19.1 ± 2.3 |
| LLM-based | ReAct(@1M) | 23.6 ± 2.7 |
| | Reflexion(@1M) | 27.7 ± 2.3 |
| | ExpeL(@1M) | 26.4 ± 2.3 |
| | KnowSelf(@1M) | 33.2 ± 2.7 |
| | GLIDER(@1M) | 31.4 ± 2.3 |
| Ours | CoS (Mistral-7B) | **39.5 ± 2.7** |

With Gemma-7B, CoS reaches 77.25% while GLIDER achieves 79.34%, showing a performance gap. This pattern contrasts with ScienceWorld where CoS shows substantial gains across both splits. The difference in improvement magnitudes reflects the varying complexity of causal reasoning required across benchmarks. ALFWorld features simple procedural tasks with short causal chains and deterministic state transitions, where the benefits of rigorous counterfactual validation are less pronounced. In contrast, ScienceWorld involves complex multi-variable interactions where systematic causal verification becomes essential for identifying effective strategies. This suggests that causal reasoning mechanisms provide greater advantages in environments with higher causal complexity. Despite modest

accuracy gains, CoS demonstrates superior step efficiency, completing tasks within 17.3–19.2 steps with a 4.5%–11.3% reduction compared to baselines' 18.7–20.4 steps, indicating that hypothesis-driven exploration remains more efficient than trial-and-error approaches.

**Performance on Crafter.** As shown in Table 2, CoS achieves 39.5% ± 2.7% accuracy, outperforming the best LLM-based baseline KnowSelf (33.2%, +19.0%) and the best RL-based method Rainbow (19.5%, +102.6%). These improvements demonstrate that systematic causal reasoning enables more effective exploration in open-ended environments. Unlike ScienceWorld and ALFWorld which provide clear task goals, Crafter requires agents to autonomously discover achievements through long-term planning. CoS addresses this challenge by proposing diverse strategies, testing them through controlled experiments, and identifying which actions lead to progress. This structured approach allows CoS to discover effective action sequences more efficiently than trial-and-error methods. The performance gains are particularly notable compared to RL-based methods, which struggle when rewards are sparse and the exploration space is large.

### 4.3. Ablation Study

**Ablation of Model Architectures.** Table 3 presents ablation results on unseen tasks across three backbones. The experiments provide two key insights into our framework's effectiveness. First, the scientific loop is essential for performance. The Hypothesis-Experiment-Validation cycle significantly improves results across all backbones. With

*Table 3.* **Training ablation on ScienceWorld (unseen).** Success rate (%) is reported across backbones and training strategies.

| Model | Configuration | SFT | CPO | SFT+CPO |
|-------|---------------|-----|-----|---------|
| Mistral-7B | w/ CoS | 62.34 | 65.18 | **67.23** |
| | w/o CoS | 51.23 | 53.67 | 56.41 |
| Gemma-7B | w/ CoS | 55.82 | 58.47 | **60.47** |
| | w/o CoS | 45.68 | 48.52 | 51.17 |
| Llama-3-8B | w/ CoS | 60.47 | 63.28 | **65.35** |
| | w/o CoS | 49.76 | 52.58 | 55.23 |

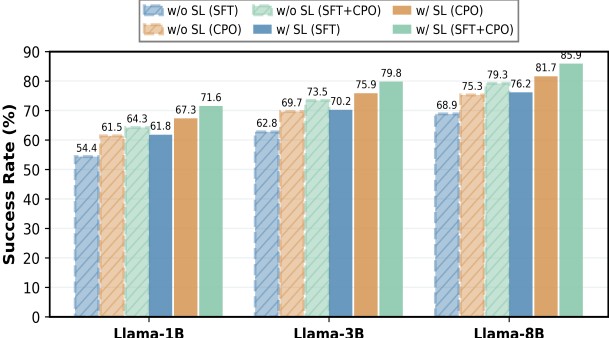

*Figure 3.* Ablations across model scales on ScienceWorld unseen tasks. We report success rate (%). Hatched bars represent models w/o Scientific Loop; solid bars represent the full CoS framework w/ Scientific Loop. Colors denote training strategies: blue for SFT only, orange for CPO only, and green for SFT+CPO.

CoS, the full SFT+CPO training achieves substantial gains over the baseline without CoS: +19.2% for Mistral-7B (67.23% vs. 56.41%), +18.3% for Llama-3-8B (65.35% vs. 55.23%), and +18.2% for Gemma-7B (60.47% vs. 51.17%). These consistent improvements demonstrate that systematic causal reasoning through controlled experimentation enables more robust decision-making than pattern memorization alone. Second, SFT and CPO exhibit synergistic effects. The SFT+CPO combination consistently outperforms single-stage training by 3-6 percentage points across all models. Interestingly, CPO-only training performs comparably to or even surpasses SFT-only training, achieving 65.18% vs. 62.34% on Mistral-7B. This confirms that Counterfactual Preference Optimization effectively reinforces causal reasoning by rewarding action chains that isolate true causal factors while penalizing spurious correlations. Consistent with standard RLHF practices, initializing CPO from SFT parameters yields the best performance, indicating that structured reasoning provides a strong foundation for preference optimization. Although all backbones follow these patterns, Mistral-7B demonstrates slightly superior performance, likely due to its stronger instruction-following capabilities.

**Effect of Model Scales.** We further investigate the impact of

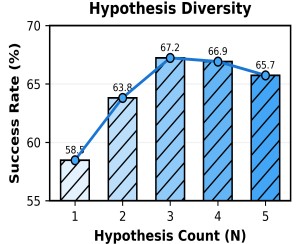
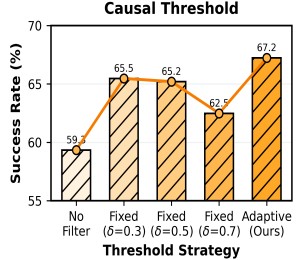

*Figure 4.* Hyperparameter analysis on the ScienceWorld benchmark by using Mistral-7B and SFT+CPO. Left: Hypothesis diversity $N$ with varying number. Right: Fixed thresholds with different $\delta$ vs. our adaptive threshold mechanism by dynamically adjusting thresholds based on policy uncertainty.

model scales in Figure 3. By evaluating Llama-family models ranging from 1B to 8B parameters on ScienceWorld's unseen tasks, we observe a consistent improvement of our framework regardless of scale. Notably, the scientific loop demonstrates remarkable efficiency even with small parameter number in Llama-1B. In the SFT+CPO setting, the Llama-3B model achieves an accuracy of 79.82%, outperforming the larger Llama-8B model trained with standard SFT (68.92%). As parameter increases, our model exhibits a consistent improvement in performance. This suggests that the causal reasoning approach effectively enhances agent capacity without large parameter counts, making the framework more practical for resource-constrained scenarios.

### 4.4. More Design Analysis

**Hypothesis Diversity Analysis.** Since the hypothesis diversity is key to the success of the proposed method, we conduct experiments to evaluate the impact of hypothesis number $N$. As shown in Figure 4 (left), model performance improves as the number of hypotheses increases, peaking at 67.23% on ScienceWorld unseen tasks when $N = 3$. At $N = 1$, the performance is limited (58.47%) due to insufficient coverage of the causal hypothesis space. If the initial conjecture is flawed, the subsequent reasoning chain fails without a recovery mechanism. However, $N \geq 4$ performance begins to decline (66.92% at $N = 4$ and 65.74% at $N = 5$), suggesting that excessive hypotheses introduce noise and increase the cognitive load on the Intervention Planner and Effect Validator. $N = 3$ strikes the optimal balance between hypothesis coverage and reasoning precision.

**Adaptive Threshold Mechanism.** Figure 4 (right) compares the adaptive threshold calibration against fixed threshold baselines. No filtering, agents confound correlation with causality without explicit intervention validation, results in a performance of only 59.34%. Fixed thresholds exhibit an inverted-U relationship: a small threshold $\delta = 0.3$ yields 65.47%, while a strict threshold $\delta = 0.7$ drops to 62.48% due to the over-rejection of valid hypotheses. The optimal

*Table 4.* Effect of self play trajectory scale and ratio on ALFWorld. We fix human data to 5K and vary self play data from 0 to 95K.

| Training Data | Human | Self play | Total | Score (%) |
|---|---|---|---|---|
| Human only | 5K | 0 | 5K | 72.18 ± 1.8 |
| Mixed 1:1 | 5K | 5K | 10K | 74.82 ± 1.7 |
| Mixed 1:4 | 5K | 20K | 25K | 77.35 ± 1.6 |
| Ours (1:9) | 5K | 45K | 50K | **80.51 ± 1.5** |
| Mixed 1:19 | 5K | 95K | 100K | 79.94 ± 1.7 |

*Table 5.* **Task-level compute and performance on ScienceWorld (unseen).** CoS Full indicates $N=3$ and $K=3$; CoS Reduced means $N=2$ and $K=2$.

| Metric | ReAct | GLIDER | CoS Full | CoS Reduced |
|---|---|---|---|---|
| LLM Calls / Step | 1 | 2 | 24 | 16 |
| Env Steps / Step | 1 | 1 | 18 | 8 |
| Steps / Episode | 127.3 | 68.1 | **61.8** | 64.8 |
| Success (%) | 17.65 | 54.32 | **67.23** | 61.85 |

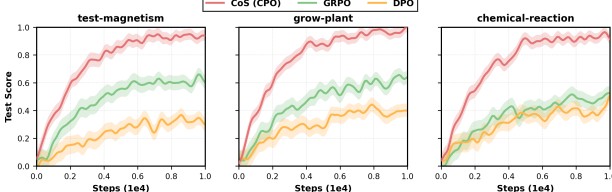

*Figure 5.* Online fine-tuning performance (success rate/100) with different preference optimization methods in ScienceWorld.

*Table 6.* **Computation and performance on ScienceWorld (unseen, Mistral 7B).** Relative performance is normalized to CoS-Full with $N=3$ and $K=3$.

| Config $(N, K)$ | Env Steps per Step | Rel. Cost | Steps / Episode | Success | Rel. Perf. |
|---|---|---|---|---|---|
| (3,3) | 18 | 100% | 61.8 | 67.23 | 100% |
| (2,3) | 12 | 67% | 63.2 | 65.47 | 97.4% |
| (3,2) | 12 | 67% | 62.4 | 64.91 | 96.6% |
| (2,2) | 8 | 44% | 64.8 | 61.85 | 92.0% |
| GLIDER | 1 | 5.6% | 68.1 | 54.32 | 80.8% |

fixed performance (65.20%) is achieved at $\delta = 0.5$. In contrast, the adaptive mechanism achieves a peak performance of 67.23% by dynamically adjusting thresholds based on policy uncertainty in Eq. (3). This improvement validates that uncertainty-driven threshold effectively balances exploration and precision: it lowers thresholds in uncertain states to explore potential hypotheses while maintaining strict filtering in confident states. Our adaptive mechanism successfully overcomes the limitations of static thresholds across diverse decision-making contexts.

**Impact of Self-Play Data.** To quantify the impact of self-play trajectories, we analyze varying proportions of human and self-play data on ALFWorld while fixing human trajectories to 5K. This clarifies how performance scales from 5K to 100K total samples, justifying our final choice of 50K samples. As shown in Table 4, augmenting the 5K human-only baseline with self-play trajectories enhances performance, confirming the efficacy of intervention-driven data augmentation. Accuracy rises from 72.18 to 80.51 as the self-play ratio reaches 1:9 (50K), but marginally declines at 1:19 (100K). This indicates a performance saturation and mild noise accumulation at larger scales. Consequently, 5K human + 45K self-play trajectories provides the optimal trade-off between data efficiency and policy quality.

**Comparison of Preference Optimization Methods.** To validate the effectiveness of our causal preference construction, we compare our method against two widely-used preference optimization baselines: Direct Preference Optimization (DPO) (Rafailov et al., 2023) and Group Relative Policy Optimization (GRPO) (Shao et al., 2024). All methods are initialized from the same checkpoint with identical computational budgets. As shown in Figure 5, our method consistently outperforms both baselines across three repre-

sentative tasks. On test-magnetism, our approach achieves 95% success rate compared to baseline methods' 35-65%. Similar trends appear on grow-plant (98% versus 42-62%) and chemical-reaction (96% versus 45-48%). Notably, our method demonstrates faster convergence, reaching over 90% performance within 5,000 training steps, while baselines plateau around 60% even after 10,000 steps. The performance gap stems from our causal preference construction strategy. Standard methods sample trajectory pairs randomly or rank them at group level, which may include spurious correlations. In contrast, our approach explicitly measures causal effects to construct preferences—trajectories with strong causal effects are labeled as positive examples, while those with weak effects are labeled as negative. Our model learns to prioritize causal verification over pattern matching, aligning with the hypothesis-validation framework.

### 4.5. Computation Overhead

We evaluate the computational overhead of CoS against representative agent baselines on the unseen split of ScienceWorld using Mistral-7B. As shown in Table 5, CoS introduces a higher compute cost per decision step due to its explicit hypothesis generation and intervention-based validation. Specifically, at $(N, K) = (3, 3)$, CoS requires 24 LLM calls and 18 environment interactions per decision step, whereas standard baselines require only 1–2 calls and a single interaction. However, this extra overhead translates into superior task-level efficiency: CoS achieves the highest success rate (67.23%) with fewer total decision steps (61.8), outperforming GLIDER (54.32% success in 68.1 steps). This underscores that step-level cost alone is insufficient to reflect end-to-end efficiency, necessitating an evaluation of

*Figure 6.* Comparison of (a) CoS, (b) ReAct, and (c) KnowSelf on ScienceWorld.

both step-wise overhead and episodic outcomes.

To further quantify the interplay between compute budget and performance, we vary the rollout configuration $(N, K)$, as shown in Table 6. We can observe that scaling up $(N, K)$ scales success at the expense of increased interactions, whereas scaled-down configurations preserve the majority of performance gains at a fraction of the cost. Notably, $(N, K) = (2, 2)$ reduces interactions per step by 56% (18→8) while retaining 92.0% of the CoS Full performance (61.85/67.23), offering a practical operating point under compute-constrained scenarios.

### 4.6. Causal Reasoning in Interactive Environments

State-of-the-art agent frameworks often fail to distinguish causation from correlation through pattern matching. As shown in Figure 6, we compare CoS against ReAct and KnowSelf on ScienceWorld and provide an example about *Making the light bulb glow*. In Figure 6(a), CoS validates causal dependencies through hypothesis generation, intervention planning, and effect validation, identifying the essential wire connections and switch operation while rejecting metal surface contact. Figure 6(b) presents ReAct's pattern-based decision to touch the metal surface, leading to spurious actions based on spatial co-occurrence rather than causal necessity. Figure 6(c) details KnowSelf's reflection, which identifies the missing power source and adds battery connection, but its knowledge retrieval incorrectly suggests metal contact because this pattern frequently appears in training data, missing the true causal requirement of switch operation. Despite adopting rethinking, correct action sequences are not produced in ReAct and KnowSelf. In contrast, CoS avoids error-prone scenarios by validating the complete causal chain. We conclude that in dynamically changing environments, an agent needs to validate causal dependencies through hypothesis-driven experimentation for robust decision making. Merely relying on pattern matching or situational knowledge retrieval is far from sufficient.

## 5. Conclusion

In this paper, we introduce the Cycle-of-Science framework for autonomous agent reasoning that shifts from heuristic-based action to a loop of Hypothesis Generation, Counterfactual Experimentation, and intervention validation. The integration of Counterfactual Preference Optimization further ensures that the agent suppresses spurious correlations in favor of verified causal drivers. We conduct extensive experiments and ablations on benchmark datasets, and results validate that our method achieves competitive performance against state-of-the-art baselines across multiple tasks.

## Impact Statement

This work improves the reliability of LLM based agents by moving from pattern matching to causal verification. Potential benefits include safer and more robust autonomous decision making in complex environments. Our method has limitations. First, CoS adds computational overhead from counterfactual rollouts at inference time. Second, failure modes remain: when all hypotheses are rejected, fallback actions can be suboptimal; under distribution shift, causal estimates may be noisy; and in long horizon tasks, context truncation can reduce consistency. Therefore, CoS should be deployed with resource aware settings and monitoring. We encourage future work on lower cost validation, better uncertainty calibration, and stronger failure handling for long horizon tasks.

## Acknowledgements

This work is supported by National Key Research and Development Program of China under Grant 2025YFA0921700, Ministry of Science and Technology of Sichuan Province Program (2024ZDZX0011& 2025ZHCG0002), the National Natural Science Foundation of China (No.62406221), the Natural Science Foundation of Tianjin (No.25JCQNJC00770).

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

# A. Algorithm

We present the Cycle-of-Science framework in Algorithm 1. The algorithm consists of three distinct stages: (1) Causal Data Construction, where we augment demonstration trajectories with causal reasoning chains through hypothesis generation, intervention design, and effect measurement; (2) Supervised Fine-Tuning (SFT), where we train the base model to generate structured causal reasoning chains; and (3) Counterfactual Preference Optimization (CPO), where we refine the policy to prioritize causally validated hypotheses over spurious correlations.

---

**Algorithm 1** Cycle-of-Science Framework

---

**Require:** Demonstration data $\mathcal{D}$demo, base model $\mathcal{M}\theta$, environment env
**Ensure:** Trained causal reasoning policy $\pi_\theta$
 1: **// Stage 1: Causal Data Construction**
 2: Initialize $\mathcal{D}c \leftarrow \emptyset$
 3: **for** each trajectory $\tau \in \mathcal{D}$demo **do**
 4:    **for** each state $h_t$ in $\tau$ **do**
 5:       Generate hypotheses $\mathcal{H}_t$ using Intuitive Reasoner $\mathcal{M}_i$ ($\tau = 0.8$)
 6:       Design interventions $\mathcal{E}_t$ via Intervention Planner
 7:       Measure causal effects $\mathcal{V}_t$ via $K = 3$ rollouts (Eq. 1)
 8:       Compute adaptive threshold $\delta_t$ (Eq. 3)
 9:       **if** $\max(\mathcal{V}_t) > \delta_t$ and reward $> 0.8$ **then**
10:          $\mathcal{D}_c \leftarrow \mathcal{D}_c \cup (h_t, \mathcal{H}_t, \mathcal{E}_t, \mathcal{V}t, a_t, ot + 1)$
11:       **end if**
12:    **end for**
13: **end for**
14: Expand $\mathcal{D}_c$ to 50K trajectories via self-play (Eq. 4)
15: **// Stage 2: Supervised Fine-Tuning**
16: Train $\pi_\theta$ to generate reasoning chains $\mathbf{c}_t = (\mathcal{H}_t, \mathcal{E}t, \mathcal{V}t)$
17: Optimize via $\mathcal{L}$SFT $= -\mathbb{E}(h_t, \mathbf{c}_t, a_t) \sim \mathcal{D}c \log \pi\theta(\mathbf{c}t, a_t|h_t)$
18: Save reference model $\pi$ref $\leftarrow \pi\theta$
19: **// Stage 3: Counterfactual Preference Optimization**
20: Construct $\mathcal{D}$pref with pairs $(h, y^+, y^-)$ where
21:    EIS$(y^+) > \delta_t$ and EIS$(y^-) \leq \delta_t$
22: Optimize via $\mathcal{L} = \mathcal{L}$CPO $+ \alpha\mathcal{L}_{\text{NLL}}$ (Eq. 6, 7)
23: **// Stage 4: Online Inference**
24: Initialize $o_0 \leftarrow$ env.reset(), $h_0 \leftarrow [o_0]$, $t \leftarrow 0$
25: **while** not done and $t < T_{\max}$ **do**
26:    Generate hypotheses $\mathcal{H}t \sim \pi\theta(\cdot|h_t, \tau = 0.8)$
27:    Design interventions $\mathcal{E}t$ and measure EIS$i$ (Eq. 1)
28:    Compute $\delta_t$ (Eq. 3) and validate hypotheses
29:    Select $a_t \leftarrow C_i :$ EIS$i > \delta_t$EIS$i$
30:    Execute $ot + 1, r_t,$ done $\leftarrow$ env.step$(a_t)$
31:    Update $ht + 1 \leftarrow h_t \cup a_t, ot + 1, t \leftarrow t + 1$
32: **end while**
33: **return** $\pi_\theta$

---

# B. Dataset

## B.1. Benchmarks

We evaluate CoS on three interactive decision-making benchmarks with distinct characteristics:

**ScienceWorld (Wang et al., 2022)** is a text-based scientific simulation environment featuring 30 tasks across 10 categories (physics, chemistry, biology). The environment provides dense rewards $\in [0, 1]$ at each step, measuring progress toward scientific objectives. Tasks require understanding of causal mechanisms in scientific domains, such as determining factors

affecting plant growth or conducting chemical reactions. The benchmark includes both seen and unseen test splits to evaluate generalization capabilities.

**ALFWorld (Shridhar et al., 2021)**   simulates household environments for navigation and object manipulation. The environment provides sparse binary rewards ($\{0, 1\}$), making exploration challenging. Tasks are categorized into six domains: pick & place, clean, heat, cool, examine, and pick two objects. Similar to ScienceWorld, it includes Text-Seen and Text-Unseen splits, where the unseen split contains out-of-distribution task variations for robustness evaluation.

**Crafter (Hafner, 2022)**   is a 2D survival environment with partial observability ($64 \times 64$ grid, $9 \times 7$ view window). The environment contains diverse objects (trees, stone, iron, coal) and hostile entities (zombies). Agents must discover and complete 22 achievements organized in a hierarchical dependency tree, e.g., collecting wood enables crafting tools, which enables mining stone. Success rate is reported by the percentage of achievements unlocked within 1M environment steps.

Table 7 presents the statistical information of our datasets for ScienceWorld and ALFWorld.

*Table 7.* Dataset statistics for ScienceWorld and ALFWorld.

| Dataset | Train | Text-Seen | Text-Unseen |
|---|---|---|---|
| ScienceWorld | 1,483 | 194 | 211 |
| ALFWorld | 3,119 | 140 | 134 |

For Crafter, we follow the standard evaluation protocol of prior work (Hafner, 2022) with 1M environment steps.

**B.2. Dataset Construction**

We construct the causal reasoning dataset $\mathcal{D}_c$ through annotation, which augments expert demonstrations with explicit causal reasoning chains. Each task starts with 100 trajectories for ScienceWorld/ALFWorld and 500 for Crafter that are processed through our four-step Hypothesis-Experiment-Validation protocol. At each state $s_t$, the Intuitive Reasoner $\mathcal{M}_i$ ($\tau = 0.8$) generates $N = 3$ hypotheses $\mathcal{H}_t = \{(C_i, E_i)\}_{i=1}^3$ pairing actions with expected effects. Following, the Intervention Planner designs counterfactual experiments $\mathcal{E}_t = \{(\text{do}(X_i), E_{cf}^i)\}_{i=1}^3$ to isolate variables. We then measure causal effects via $K = 3$ rollouts using Eq. (1). Finally, hypotheses are validated as $y^+$ when $\text{EIS}_i > \delta_t$ or rejected as $y^-$ otherwise.

To illustrate this process in detail, consider a ScienceWorld's plant growth task where the state features a wilted plant, watering can, fertilizer, and grow light. In Step 1 (Hypothesis Generation), we generate three hypotheses: $H_1$ proposes using fertilizer with the expectation that nutrients aid growth, $H_2$ suggests adding water as essential for survival, and $H_3$ advocates activating the grow light to enable photosynthesis. In Step 2 (Intervention Design), we construct counterfactual experiments comparing do(fertilizer) vs. baseline, do(water) vs. baseline, and do(light) vs. baseline. In Step 3 (Effect Measurement), controlled rollouts yield $\text{EIS}_1 = 0.15$ (fertilizer alone proves insufficient without water), $\text{EIS}_2 = 0.73$ (immediate recovery observed with water), and $\text{EIS}_3 = 0.58$ (light is essential but shows slower impact than water). In Step 4 (Validation), with threshold $\delta_t = 0.5$, we validate $H_2$ and $H_3$ as causally effective ($y^+$) while rejecting $H_1$ as a spurious correlation ($y^-$) since its effect falls below the threshold. Quality filtering retains only trajectories with reward $> 0.8$, yielding 5K seed demonstrations that are expanded to 50K via self-play (see Table 9). For CPO, we sample $M = 5$ chains per state from $\pi_{\text{ref}}$ and select pairs ($y^+, y^-$) with EIS margin $> 0.3$, yielding 200K preference pairs summarized in Table 8.

*Table 8.* **Final dataset composition and causal effect statistics.** EIS values represent mean $\pm$ standard deviation for validated ($y^+$) and rejected ($y^-$) hypotheses across benchmarks.

| Benchmark | Trajectories | Steps | CPO Pairs | EIS ($y^+$) | EIS ($y^-$) |
|---|---|---|---|---|---|
| ScienceWorld | 20,000 | 374,000 | 80,000 | $0.71 \pm 0.18$ | $0.12 \pm 0.08$ |
| ALFWorld | 24,500 | 301,000 | 90,000 | $0.23 \pm 0.12$ | $0.08 \pm 0.05$ |
| Crafter | 5,500 | 234,000 | 30,000 | $0.54 \pm 0.21$ | $0.15 \pm 0.09$ |
| **Total** | **50,000** | **909,000** | **200,000** | – | – |

**Self-Play Data Scaling.**   To characterize the impact of self-play trajectories on model quality, we conduct a scaling study by varying the human-to-self-play ratio while keeping the human seed set fixed at 5K demonstrations (i.e., 100 trajectories

*Table 9.* **Impact of self-play data scaling on ALFWorld (Mistral-7B).** Performance (mean $\pm$ std) across different human-to-self-play ratios with a fixed 5K human seed set.

| Data Mix | Human | Self-play | Total | Score (%) |
|---|---|---|---|---|
| Human only | 5K | 0 | 5K | $72.18 \pm 1.8$ |
| Mixed 1:1 | 5K | 5K | 10K | $74.82 \pm 1.7$ |
| Mixed 1:4 | 5K | 20K | 25K | $77.35 \pm 1.6$ |
| **Ours (1:9)** | **5K** | **45K** | **50K** | $\mathbf{80.51 \pm 1.5}$ |
| Mixed 1:19 | 5K | 95K | 100K | $79.94 \pm 1.7$ |

per task aggregated across benchmarks). As shown in Table 9, performance on ALFWorld improves monotonically as the proportion of self-play data increases, peaking at the 1:9 ratio (50K total, $80.51 \pm 1.5\%$). Beyond this point (1:19 ratio, 100K total), performance saturates at $79.94 \pm 1.7\%$, confirming that the 50K scale provides an optimal trade-off between data efficacy and computational cost.

This augmentation process transforms simple state-action demonstrations into rich causal reasoning trajectories, enabling the model to learn not just *what* actions to take, but *why* those actions are causally effective.

*Table 10.* Model configuration and LoRA settings.

| Model | Parameters | Reference |
|---|---|---|
| Mistral-7B-Instruct-v0.2 | 7B | Jiang et al. (2023) |
| Gemma-7B-Instruct | 7B | Gemma Team et al. (2024) |
| Llama-3-8B-Instruct | 8B | Dubey et al. (2024) |
| *LoRA Configuration* | | |
| Rank ($r$) | | 16 |
| Alpha ($\alpha$) | | 32 |
| Target Modules | | q_proj, v_proj, k_proj, o_proj |

*Table 11.* Training hyperparameters for SFT and CPO.

| Hyperparameter | SFT | CPO |
|---|---|---|
| Learning Rate | $5 \times 10^{-5}$ | $1 \times 10^{-5}$ |
| LR Scheduler | Cosine Decay | Cosine Decay |
| Batch Size | 32 | 16 |
| Gradient Accumulation | 4 steps | 8 steps |
| Training Epochs | 3 | 2 |
| Max Sequence Length | 2048 tokens | 2048 tokens |
| Optimizer | AdamW | AdamW |
| $\beta_1, \beta_2$ | 0.9, 0.999 | 0.9, 0.999 |
| Warmup Ratio | 0.1 | 0.1 |
| Weight Decay | 0.01 | 0.01 |
| *CPO-Specific Parameters* | | |
| KL Penalty ($\beta$) | - | 0.1 |
| Length Regularization ($\alpha$) | - | 0.01 |

# C. Implementation Details

## C.1. Model Configuration

We implement CoS using three open-source language models as backbones shown in Table 10. All models are fine-tuned using LoRA to reduce computational costs while maintaining performance.

## C.2. Training Hyperparameters

Table 11 summarizes the hyperparameters used for both supervised fine-tuning (SFT) and counterfactual preference optimization (CPO) stages.

## C.3. Inference Configuration

Table 12 presents the inference-time hyperparameters during agent deployment.

*Table 12.* Inference configuration for online decision-making.

| Parameter | Value |
|---|---|
| Hypothesis Generation Temperature ($\tau$) | 0.8 |
| Action Selection Temperature ($\tau$) | 0.1 |
| Hypothesis Diversity ($N$) | 3 |
| Rollout Samples per Hypothesis ($K$) | 3 |
| Baseline Threshold ($\delta_{\text{base}}$) | 0.5 |
| Adaptive Modulation Coefficient ($\lambda$) | 0.3 |
| *Environment-Specific* | |
| $T_{\max}$ (ScienceWorld/ALFWorld) | 50 steps |
| $T_{\max}$ (Crafter) | $10^6$ steps |

## C.4. Environment-Specific Settings

Table 13 details the characteristics of each benchmark environment.

*Table 13.* Environment-specific configurations.

| Attribute | ScienceWorld | ALFWorld | Crafter |
|---|---|---|---|
| Action Space Size | $\sim$100 | $\sim$20-30 | 17 |
| Observation Type | Natural Language | Natural Language | Text Description |
| Avg. Obs. Tokens | 150 | 80 | 100 |
| Reward Type | Dense $[0, 1]$ | Sparse $\{0, 1\}$ | Achievement-based |
| Reward Frequency | Every step | At completion | Per achievement |
| Partial Observability | No | No | Yes ($9 \times 7$ view) |
| Grid Size | - | - | $64 \times 64$ |

# D. Additional Ablations on Crafter

To provide deeper insights into CoS's effectiveness on open-ended exploration tasks, we conduct comprehensive ablation studies on Crafter. Unlike ScienceWorld and ALFWorld with clear task objectives, Crafter requires autonomous achievement discovery through long term planning, making it an ideal testbed for evaluating causal reasoning under sparse rewards.

## D.1. Component Ablation

Table 14 presents ablation results validating the necessity of each component in the Cycle-of-Science framework.

*Table 14.* **Impact of core components on Crafter.** Scores represent percentage of achievements unlocked (mean ± std over 3 runs, 1M steps each).

| Configuration | Score (%) |
|---|---|
| *Full Framework* | |
| CoS (Full) | **39.5 ± 2.7** |
| *Remove Individual Components* | |
| w/o Causal Hypothesis Generation | 28.3 ± 2.5 |
| w/o Intervention Planning | 31.7 ± 2.4 |
| w/o Effect Validation | 26.9 ± 2.6 |
| *Remove Training Stages* | |
| SFT only (no CPO) | 34.8 ± 2.3 |
| CPO only (no SFT) | 30.2 ± 2.8 |
| No Scientific Loop (baseline SFT) | 25.6 ± 2.5 |

**Analysis of Component Contributions.** Removing any one component leads to substantial performance degradation:

- **w/o Causal Hypothesis Generation (-11.2%):** Without diverse hypothesis generation ($N = 1$ greedy), the agent fails to explore multi-step achievement dependencies. For example, it may collect wood repeatedly without hypothesizing tool crafting, missing critical causal chains like wood $\rightarrow$ table $\rightarrow$ tools $\rightarrow$ mining.

- **w/o Intervention Planning (-7.8%):** Without counterfactual experiment design, the agent cannot isolate which actions causally contribute to achievement unlocks. It confuses coincidental successes (e.g., zombie defeat while exploring) with intentional strategies (e.g., crafting sword for combat).

- **w/o Effect Validation (-12.6%):** This shows the largest drop, confirming that intervention validation is critical for filtering spurious correlations in sparse reward settings. Without validation, the agent pursues misleading patterns like "moving near iron unlocks achievements" (correlation) rather than "mining iron after crafting pickaxe unlocks achievements" (causation).

The training stage ablations reveal complementary effects: SFT alone (34.8%) establishes basic causal reasoning patterns from demonstrations, while CPO only (30.2%) struggles without proper initialization. The combination (39.5%) achieves synergistic gains. Notably, removing the entire scientific loop reduces performance to 25.6%, demonstrating that standard supervised learning on demonstrations is insufficient for open-ended exploration—agents must actively discover and validate causal mechanisms.

### D.2. Adaptive Threshold Mechanism

Table 15 evaluates the impact of our adaptive threshold calibration in Eq. (3) compared to fixed validation criteria.

*Table 15.* **Impact of adaptive threshold mechanism on Crafter.** We compare fixed thresholds against dynamic calibration based on policy entropy.

| Threshold Strategy | Score (%) |
|---|---|
| No Validation (all hypotheses accepted) | 22.4 ± 2.9 |
| Fixed $\delta = 0.3$ | 33.7 ± 2.6 |
| Fixed $\delta = 0.5$ | 35.8 ± 2.4 |
| Fixed $\delta = 0.7$ | 31.2 ± 2.5 |
| **Adaptive $\delta_t$ (Ours)** | **39.5 ± 2.7** |

**Why Adaptive Thresholds Matter?** Without validation (22.4%), the agent wastes effort on spurious correlations. For example, an agent repeatedly attacks trees that is correlated with wood gathering without crafting planks (causal prerequisite

for progression). Moreover, fixed thresholds present an inverted-U relationship, where low thresholds ($\delta = 0.3$, 33.7%) accept false positives like "moving randomly helps" (high EIS due to coincidental resource encounters), while high thresholds ($\delta = 0.7$, 31.2%) over-filter valid exploratory actions with moderate EIS such as "explore new biomes to find iron."

The optimal fixed threshold ($\delta = 0.5$, 35.8%) still underperforms our adaptive mechanism (39.5%) by 3.7%. The adaptive strategy dynamically adjusts validation strictness based on policy uncertainty in Eq. (2):

- **Early exploration (high entropy):** The agent faces many unknown achievement paths, e.g., "Should I prioritize mining or farming?". Low $\delta_t$ encourages testing diverse hypotheses, enabling discovery of multi-branch progression trees.

- **Late exploitation (low entropy):** Once a promising strategy emerges, e.g., "Craft iron tools to mine coal", high $\delta_t$ commits to validated causal chains, avoiding distraction by weakly correlated actions.

This entropy driven modulation is crucial for Crafter's hierarchical achievement structure, where exploration-exploitation trade-offs shift dramatically as the agent progresses from basic survival to complex crafting chains.

### D.3. Hypothesis Diversity Analysis

Table 16 examines the trade-off between hypothesis diversity $N$ and computational efficiency.

*Table 16.* **Impact of hypothesis diversity $N$ on Crafter.** We vary the number of hypotheses generated per decision step.

| $N$ | Score (%) | Time per Episode (s) |
|---|---|---|
| $N = 1$ | $30.2 \pm 2.8$ | 84.3 |
| $N = 3$ (Ours) | $\mathbf{39.5 \pm 2.7}$ | 152.6 |
| $N = 5$ | $38.9 \pm 2.5$ | 241.8 |
| $N = 10$ | $37.1 \pm 2.9$ | 478.5 |

**Optimal Hypothesis Count.** With $N = 1$ (30.2%), the agent lacks exploration breadth and frequently gets trapped in local optima. For example, it may focus solely on survival actions (collecting food, fleeing zombies) without hypothesizing achievement-oriented strategies like tool crafting or dungeon exploration. This myopic behavior limits progress beyond basic survival achievements (3-5 out of 22 total).

Increasing to $N = 3$ (39.5%) provides sufficient coverage of Crafter's action space with reasonable overhead (152.6s per episode, 1.8× baseline). Three hypotheses typically span orthogonal strategies: (1) resource gathering, (2) crafting progression, (3) combat/exploration. This diversity enables the validation mechanism to identify the causally effective path.

Beyond $N = 3$, performance plateaus: $N = 5$ (38.9%) and $N = 10$ (37.1%) show diminishing returns despite 1.6× and 3.1× longer inference. Two factors explain this:

1. **Hypothesis redundancy:** Additional hypotheses often duplicate existing strategies with minor variations, e.g., "mine iron at location A" vs. "mine iron at location B", failing to explore genuinely distinct causal pathways.

2. **Validation noise:** More hypotheses increase the risk of spurious high-EIS measurements due to environmental stochasticity, e.g., lucky zombie spawns during rollouts, which the threshold mechanism struggles to filter completely.

The optimal $N = 3$ balances exploration thoroughness with computational efficiency, capturing the key causal branches in Crafter's achievement tree without exhaustive enumeration.

### D.4. Rollout Sample Size

Table 17 evaluates the impact of rollout samples $K$ on causal effect estimation accuracy in Eq. (**??**).

*Table 17.* **Impact of rollout samples $K$ on Crafter.** We vary the number of environment rollouts used for EIS measurement.

| $K$ | Score (%) | Total Interactions |
|---|---|---|
| $K = 1$ | $33.8 \pm 3.1$ | $3.2 \times 10^6$ |
| $K = 3$ (Ours) | $\mathbf{39.5 \pm 2.7}$ | $5.7 \times 10^6$ |
| $K = 5$ | $39.8 \pm 2.4$ | $8.9 \times 10^6$ |
| $K = 10$ | $40.1 \pm 2.6$ | $16.4 \times 10^6$ |

**Balancing Statistical Reliability and Sample Efficiency.** With $K = 1$ (33.8%), causal measurements are noisy due to Crafter's stochastic dynamics—zombie spawns, resource availability, and weather conditions vary across trials. Single-sample EIS estimates often misjudge causality: for instance, a hypothesis "explore caves" may show low EIS if a zombie ambush occurs in that specific rollout, even though cave exploration is genuinely beneficial for finding iron.

Increasing to $K = 3$ (39.5%) stabilizes EIS estimates through averaging, reducing false negatives (rejecting valid hypotheses due to unlucky rollouts) with moderate cost (5.7M interactions total). The standard deviation also decreases from $\pm 3.1$ to $\pm 2.7$, indicating more consistent agent behavior.

Further increasing $K$ yields marginal gains: $K = 5$ (39.8%, +0.3%) and $K = 10$ (40.1%, +0.6%) improve robustness slightly but require 1.6× and 2.9× more interactions. Given Crafter's 1M-step evaluation budget, exhaustive rollouts become prohibitively expensive. The $K = 3$ setting provides near-optimal performance while remaining computationally feasible, suggesting this configuration generalizes well to other sparse-reward environments where extensive interaction is costly.

## E. Prompts for Causal Reasoning

This section provides the complete prompt templates used in our Cycle-of-Science framework. All prompts are designed to elicit structured causal reasoning from language models, following the Hypothesis-Experiment-Validation loop described in Section 3.2.

The Hypothesis Generator employs high-temperature sampling ($\tau = 0.8$) to encourage diverse exploratory reasoning, while the Analytical Reasoner uses low-temperature sampling ($\tau = 0.1$) for rigorous evaluation. The prompt structures are illustrated in Figure 7.

To demonstrate the complementary roles of the two reasoners, we present their interactions in Figure 7 using the plant growth task from ScienceWorld. The highlighted portions reveal two critical aspects of our causal reasoning framework: *divergent hypothesis generation* (left panel, highlighted in blue) where the Intuitive Reasoner explores multiple plausible causal pathways, and *convergent effect validation* (right panel, highlighted in orange) where the Analytical Reasoner rigorously quantifies causal impacts through controlled experiments.

### E.1. Intuitive Reasoner: Exploratory Hypothesis Generation

The Intuitive Reasoner (left panel) operates in an exploratory mode with high temperature ($\tau = 0.8$) to maximize hypothesis diversity. Given the current state—a greenhouse containing a wilted plant, watering can, fertilizer, and grow light—the reasoner generates three distinct hypotheses targeting different causal mechanisms:

**Hypothesis 1** proposes using fertilizer, hypothesizing that nutrient deficiency is the primary cause of wilting. This reflects a common agricultural intervention pattern where fertilizer application addresses growth limitations.

**Hypothesis 2** suggests watering the plant, targeting hydration as the critical factor. This hypothesis is grounded in the fundamental biological requirement for water in cellular function and turgor pressure maintenance.

**Hypothesis 3** advocates activating the grow light, focusing on photosynthetic capacity as the limiting factor. This addresses the energy production pathway essential for plant metabolism.

The diversity across these hypotheses is crucial: they target distinct causal pathways (nutrient supply, hydration, energy production) rather than redundant variations of the same mechanism. This exploratory breadth ensures comprehensive coverage of the causal hypothesis space, preventing premature convergence on a single explanation that may prove spurious upon validation.

## Prompts

**Intuitive Reasoner Prompt:**
You are an intuitive scientific reasoner. Based on the current state and goal, generate diverse causal hypotheses about which actions might achieve the objective.
**Input:**
Task Goal: [goal description]
Current Observation: [state description]
Action History: [previous actions]
Available Actions: [action space]
**Task:** Generate N=3 distinct hypotheses, each pairing a candidate action with its expected causal effect.

**Analytical Reasoner Prompt:**
You are a rigorous analytical evaluator. Based on experimental rollout data, objectively measure the causal effect of each intervention.
**Input:**
Hypothesis: [hypothesis statement]
Initial State: [state description]
Intervention Outcomes (K=3): [state after intervention, trial 1/2/3]
Baseline Outcomes (K=3): [state without intervention, trial 1/2/3]
Expected Outcome: [what should happen if hypothesis is correct]
**Task:** Score each outcome (0-1 scale) based on progress toward the expected effect, then compute Average Causal Effect (ACE).

*Figure 7.* **Prompt templates for the Intuitive and Analytical Reasoners.** (Left) The Intuitive Reasoner with high-temperature sampling ($\tau = 0.8$) generates diverse causal hypotheses. (Right) The Analytical Reasoner with low-temperature sampling ($\tau = 0.1$) rigorously evaluates causal effects through counterfactual comparisons.

### E.2. Analytical Reasoner: Rigorous Effect Measurement

The Analytical Reasoner (right panel) employs low-temperature sampling ($\tau = 0.1$) to ensure objective, consistent evaluation. For each hypothesis generated by the Intuitive Reasoner, the Analytical Reasoner conducts controlled counterfactual experiments by comparing intervention outcomes against baseline conditions. Consider Hypothesis 2 (water intervention) illustrated in the right panel. The reasoner performs $K = 3$ independent rollouts under intervention (watering applied) and baseline (no watering) conditions from identical initial states. Each rollout is scored on a 0-1 scale based on observable progress toward the expected outcome (turgor restoration):

**Intervention trials** consistently show strong effects: Trial 1 exhibits full turgor restoration (score: 0.8), while Trials 2 and 3 demonstrate clear reversal of wilting (scores: 0.7). The consistency across trials mean intervention score of 0.73 , indicates a robust causal relationship.

**Baseline trials** uniformly yield zero scores, as the plant remains wilted without intervention. This stark contrast provides the counterfactual evidence necessary for causal inference. The computed Average Causal Effect (EIS = 0.73 - 0.00 = 0.73) quantifies the magnitude of water's causal impact. By repeating this process for all three hypotheses, the Analytical Reasoner produces a ranked ordering: water (EIS = 0.73) over light (EIS = 0.58) over fertilizer (EIS = 0.15), revealing that while all factors contribute to plant health, water addresses the most immediate causal bottleneck.

