# OpenReview forum: "Cycle-of-Science: Reliable Reasoning through Counterfactual Verification for Agent Decision Making"
_ICML.cc/2026/Conference — ICML 2026 regular_

### Official Review · Reviewer_Epb5 · 2026-03-08

**Soundness:** 2
**Presentation:** 2
**Significance:** 3
**Originality:** 3
**Overall Recommendation:** 4
**Confidence:** 4

**Summary:**

This paper proposes an approach for LLM-based agent decision-making structured around an iterative hypothesis-experiment-validation loop inspired by the scientific method. The approach is known as Cycle of Science (CoS) and works by generating causal hypotheses, designing what are referred to as “interventions”, and validating them via average causal effect scoring with adaptive thresholds. The method involves a two-stage pipeline that finetunes a base LLM by combining supervised fine-tuning with counterfactual preference optimization (CPO) to instill causal reasoning. Experiments on some datasets (ScienceWorld, ALFWorld, and Crafter) demonstrate improvements over baselines.

**Compliance With Llm Reviewing Policy:**

Affirmed.

**Final Justification:**

I thank the authors for their helpful clarifications and willingness to adopt some of my suggested changes. I have increased my score accordingly.

**Key Questions For Authors:**

Some comments and questions follow:

I found the abstract imprecise and hard to follow. For instance, it is unclear what variables are being referenced when “causal dependencies” are mentioned. There are some terms that I feel are too vague for an abstract, like “causal validation” and “policy uncertainty”. I recommend keeping the abstract language simple and clear.

The first line mentions LLMs as a “cognitive engine of interactive agents”. What does this mean?

In the contributions listed as bullets on page 2, the second bullet is the same as the first one. I recommend rewriting the contributions, possibly making the empirical study as a separate contribution.

Is the high-level intent E_i mentioned on page 3 in textual form? If so, please specify. It would also help to show examples (perhaps I missed them).

I don’t understand the indexing in equation (4). Earlier, H_t was shown to be a set of N hypotheses, and then there is a union over N in equation (4). Something seems wrong here.

Is the counterfactual mentioned in IP really a counterfactual? Or is it an intervention? Since there is no connection to a formal causal model, the language here feels imprecise. Please refer to my earlier comment about connecting to more formal causal modeling terminology.

The literature review section on causal reasoning (2.2) does not seem suitable for the work; many of the papers cited here have a very different flavor from the work in this paper, and I don’t see how they are relevant. There is a reference Chen et al. (2025) – which is referred to as Ban et al. – that one seems important. In fact, it seems to be the only truly relevant reference. Please explain how the proposed work is different.

The authors claim that equation (1) is the standard definition of ACE but I’m not sure this is true, given that there does not appear to be a formal underlying causal model.

Why is K as low as 3? That feels like an insufficient number of rollouts.

I did not fully follow the approach described for adaptive threshold modulation – it seems to be motivated by some prior work. I would find a brief explanation helpful.

The experimental investigation is reasonably thorough, but it would have been useful to gauge how the method performs with larger LLMs. I gather that smaller models were used because they needed to be hosted locally or on a single GPU. Please describe the setup (and any constraints), if possible.

The gains on ALFWorld seen tasks in the experiments are inconsistent — CoS underperforms GLIDER with Gemma-7B. How can this be explained?

There are numerous typos and grammatical issues in the paper, which I hope the authors can fix. Here are a few places I noted: “Despite effective” (abstract, page 1), “To achieve this goal, …” (page 1), “when encountering” (page 1), “an ideal decision-making” (page 2), “growt” (page 2), incorrect start quotation marks (page 3), etc.

**Limitations:**

A deeper limitations section would be beneficial.

**Strengths And Weaknesses:**

Strengths:

In my view, the primary strength of the paper lies in the empirical evaluation. Results on ScienceWorld are particularly compelling, with CoS achieving reasonable gains over the strongest fine-tuning baseline (GLIDER) on seen tasks, and comparable gains on unseen tasks across all three backbone models, suggesting the improvements are not model-specific. I also found the ablation studies to be thorough: the authors isolate the contributions of the individual components and examine various other effects (like hypothesis diversity N and rollout count K). Computational cost analysis in the appendix is a useful addition.

Another major strength is that some aspects of the work are novel. For instance, the proposed CPO stage is noteworthy as it constructs preference pairs from empirically measured difference effects rather than relying on human preferences or heuristic trajectory rankings. This is a reasonable and scalable alternative to standard DPO or GRPO.

Weaknesses:

In my view, a major weakness of the paper is the inappropriate usage of causal language. Several causal concepts are mentioned explicitly but never formalized. It is unclear how deeply the authors understand causal concepts. For instance, “do” interventions are mentioned but these are defined on causal networks, which are never constructed. Another example is the ACE computation in equation (1), which compares outcomes under intervention versus a no-intervention baseline across K=3 rollouts evaluated by an LLM scorer. This is not average causal effect estimation in any formal sense: there is no causal graph, structural equations, or any mention of confounding. As a result, the central claim that the framework performs genuine causal reasoning is not well-supported formally.  The paper would have been stronger without the causal story. For instance, ACE could be repositioned without “strong” causal language, and the overall approach could be seen as a structured hypothesis-testing heuristic. The alternative is to properly ground the framework to causality.

Another major issue is the writing: I found several issues with the paper’s exposition. Numerous grammatical errors and a literature review that is not ideal make the paper hard to follow in several places. The paper could improve considerably with some heavy editing.

Overall, I think this paper has novel ideas and shows some solid experimental results. However, I think the weaknesses balance the strengths. I am open to revising my opinion of the paper based on the discussion period.

---

> ### Author Rebuttal · Authors · 2026-03-31
>
> > **`W1: Causal terminology is not used rigorously enough`**
>
> We appreciate this critical feedback.  Our method is an intervention-based hypothesis-testing mechanism rather than a structural causal discovery framework. Specifically, CoS is designed to distinguish causal effectiveness from spurious correlations by evaluating the impact of candidate actions through controlled "do-interventions.".  To address the reviewer's concern and avoid overclaiming, we will revise the terminology throughout the paper, including replacing broad causal terms with intervention-based reasoning or causally inspired hypothesis validation, and explicitly state that our focus is on local action-effect verification rather than full causal structure recovery.
>
> > **`Key Question 1: Some terms in the abstract are too abstract`**
>
> We agree that the abstract would benefit from more concrete language. We will revise abstract terminology with formal definitions. For example, **“policy uncertainty”** will be redefined as the entropy of the action distribution, and other abstract descriptors will be rewritten to directly reflect their algorithmic implementations.
>
> > **`Key Question 2: Wording in the introduction`**
>
> We appreciate this suggestion for clarity. Our intent is that LLMs serve as the core reasoning and decision component in interactive agents. In the revision, we will replace this with: “LLMs as the reasoning and decision module of interactive agents.”
>
> > **`Key Question 3: The contribution bullets are repetitive and not clearly structured`**
>
> To improve clarity, we will reorganize our contributions into three aspects: (1) the CoS method, (2) the preference-data construction and CPO training pipeline, and (3) the empirical study across the three benchmarks.
>
> > **`Key Question 4: The notation for E_i and the indexing in Eq. (4) are unclear`**
>
> In our manuscript, $E_i$ represents a **textual high-level effect description****,** specifying the intended state change of a candidate action, and we will add illustrativeexamples in the revision.  We acknowledge the notation reuse in Eq. (4), where $N$ refers to both the hypothesis count and the sample dimension. We will rectify this by using distinct symbols.
>
> > **`Key Question 5: The related work on causal reasoning is not sufficiently connected to this paper`**
>
> We thank the reviewer for helpful suggestions! We will compare our framework with Chen et al. (2025) to clarify that Chen et al. (2025)  focuses on causal structure learning from interaction histories, whereas our work is designed for **rollout-based hypothesis testing at decision time**. Furthermore, we will correct the citation inconsistency you noted.
>
> > **`Key Question 6: Why is K=3 sufficient?`**
>
> The choice of $K$=3 represents a trade-off between stability and computational cost. Since CoS compares multiple hypotheses at each step, the total online inference cost scales linearly with $K$. As shown in the appendix, increasing $K$ from 1 to 3 yields substantial improvement gains, however, further scaling to 5 or 10 results in marginal gains while significantly increasing the computational overhead. Thus, $K=3$ serves as an empirical balance that provides sufficient hypothesis diversity for robust causal filtering without incurring prohibitive inference costs.
>
> > **`Key Question 7: The intuition behind the adaptive threshold is not clear enough`**
>
> The key intuition is that policy entropy indicates decision ambiguity: when the policy is uncertain, we relax the threshold to avoid pruning of potentially valid hypotheses that might be filtered out due to noise; when it is more certain, we utilize a stricter threshold to suppress low-quality candidates and maintain high-precision causal verification. By dynamically adjusting the threshold relative to policy entropy, CoS achieves a robust balance between exploration and rigorous filtering.
>
> > **`Key Question 8: Small LLM backbones used`**
>
> We use these backbones mainly to ensure alignment with existing baselines while keeping rollout-based validation computationally feasible. These models are the standard backbones used in previous agent benchmarks, allowing for a fair comparison with established baselines.  We will use larger-backbone evaluation as future work.
>
> > **`Key Question 9: The unstable gains on the ALFWorld seen split`**
>
> The Seen split tasks in ALFWorld are characterized by low causal ambiguity and rigid procedures, leading to near-identical rollout outcomes across different actions. We will conduct more limitation analyses in the revision.
>
> > **`Minor points: Grammatical errors and typos`**
>
> We have corrected the listed typos and will further polish the manuscript for grammar, clarity, and consistency.

---

> > ### Author Rebuttal · Reviewer_Epb5 · 2026-04-03
> >
> > I thank the authors for clarifications and for describing proposed changes. I agree that "intervention-based hypothesis-testing mechanism" is a more suitable framing for the proposed method. I still remain concerned about the causal language and hope the authors will adequately handle those edits. I am open to increasing my score slightly but I will wait until the discussion period.

---

> > > ### Author Response · Authors · 2026-04-07
> > >
> > > We sincerely thank the reviewer for the constructive feedback and for recognizing our effort to reframe the method. We fully appreciate the importance of terminological precision in the context of causality.
> > > To resolve the remaining concerns regarding causal language, we have implemented the following targeted revisions:
> > >
> > >
> > > **1. Renaming the ACE metric:** As you noted, our previous use of ACE (Average Causal Effect) lacks structural equations or confounder adjustments for formal causal inference. We have renamed the metric to **"Empirical Intervention Score (EIS)"** throughout the manuscript.
> > > It is defined as a heuristic measure that quantifies the divergence between outcomes against a baseline under different action-based interventions within the environment, rather than a formal causal effect.
> > >
> > >
> > > **2. Revising "do-intervention" terminology:** We have replaced references to "do-interventions" with more accurate descriptors. We will use terms such as **"controlled environment interventions"** or **"action-based heuristic tests"** to reflect the structured hypothesis-testing approach.
> > >
> > > We believe these precise adjustments address the concerns while maintaining the integrity of our experimental findings.
> > > Given the clarified framing, we would be deeply grateful if the reviewer would consider raising the score to reflect the improved clarity of the work!

---

### Official Review · Reviewer_8yR2 · 2026-03-12

**Soundness:** 4
**Presentation:** 3
**Significance:** 3
**Originality:** 3
**Overall Recommendation:** 5
**Confidence:** 3

**Summary:**

This paper proposes Cycle-of-Science (CoS), an LLM-based agent framework that structures decision-making as an iterative loop of Hypothesis Generation, Counterfactual Experimentation, and Causal Validation. The approach computes average causal effects (ACE) via controlled environment rollouts to filter spurious hypotheses with an adaptive, uncertainty-calibrated threshold, and then trains the agent through supervised fine-tuning and a Counterfactual Preference Optimization (CPO) objective that prefers causally validated reasoning chains. Experiments on ScienceWorld, ALFWorld, and Crafter report state-of-the-art performance (especially on ScienceWorld and Crafter), with extensive ablations on the scientific loop, training strategies, model scale, and hyperparameters.

**Compliance With Llm Reviewing Policy:**

Affirmed.

**Key Questions For Authors:**

1. How do you implement “resetting the environment to the same initial state” across ScienceWorld, ALFWorld, and Crafter? Do you serialize simulator state, control random seeds, or approximate via reinitialization scripts? Please detail any environment-specific limitations.
2. What is the computational overhead (rollouts per decision, wall-clock per episode) relative to baselines, and how does the adaptive threshold reduce the number of experiments in practice?
3. For ALFWorld where CoS sometimes trails GLIDER on seen splits, which task categories are most affected, and what failure modes do you observe?

**Limitations:**

yes

**Strengths And Weaknesses:**

Strengths
1. Technical novelty and innovation：
a) CoS constructs the agent's decision-making process as a three-stage "scientific method" loop, used for agent reasoning, operationalization of hypothesis enumeration, intervention testing, and effect verification in interactive environments.
b) The CPO training objective (preference learning tied to measured ACE rather than human labels or coarse outcomes) is a thoughtful design that connects environment-grounded causal tests directly with policy optimization.
c) The ACE-based causal filter, combined with an entropy-adaptive threshold, is a principled and practical mechanism to select actions that are more likely causal rather than merely correlated.
2. Experimental rigor and validation:
a) Broad evaluation across three distinct benchmarks (ScienceWorld, ALFWorld, Crafter), including both seen/unseen splits and long-horizon exploration scenarios.
b) Ablations isolate contributions of the scientific loop and training strategies (SFT vs. CPO vs. both), showing consistent benefits and synergy.
c) Additional analyses on hypothesis diversity and the adaptive threshold provide useful insights into design choices.
3. Clarity of presentation:
a) The overall system diagram and pipeline breakdown are easy to follow; the roles of the Intuitive and Analytical reasoners are well-motivated.
4. Significance of contributions:
a) Demonstrating that counterfactual verification can improve agent robustness and sample efficiency is timely and relevant as LLM agents become more widely deployed.
b) Large gains on ScienceWorld and Crafter suggest the approach is particularly valuable in tasks with richer causal structure and sparse rewards, respectively.

Weaknesses
1. Technical limitations or concerns:
a) Inconsistency in the intervention design: the text states both “invert the action (do not press)” and later “intervention executes the proposed action vs. baseline (no intervention).” This ambiguity undermines the conceptual soundness of the “counterfactual” and requires clarification about what is actually executed in each arm.
b) The feasibility of resetting to “the same initial state” for multiple rollouts is assumed. Many simulators do not expose full state cloning; the paper should detail how exact-state resets are implemented across all benchmarks.
2. Experimental gaps or methodological issues:
a) No computational cost analysis (number of rollouts per step, wall-clock/time-to-solve, memory/state checkpointing) relative to baselines; fairness of comparisons is unclear given the extra rollouts and resets.
b) On ALFWorld (lower “causal complexity”), improvements are modest and occasionally behind GLIDER for seen tasks; more diagnostic analyses would clarify conditions where CoS may underperform.
3. Clarity or presentation issues:
a) Some minor inconsistencies/typos and duplicated phrasing detract from polish; the notational switch between “inverted action” vs. “execute action” is the most serious clarity issue.

---

> ### Author Rebuttal · Authors · 2026-03-31
>
> > **`Weakness 1(a) & Weakness 3: Ambiguity in the intervention design`**
>
> We thank the reviewer for this constructive feedback. The 'press / do not press' example is intended to intuitively illustrate counterfactual intervention, namely comparing the outcome difference between applying a candidate action and not applying that action from the same state. It is **not** intended as a formal negation-based counterfactual test. In our implementation, we start from the same decision state, apply the candidate action in the intervention rollout, and compare it against a baseline rollout without that target intervention. For clarity, we will replace the phrase “invert the action” with more precise terminology.
>
> > **`Weakness 1(b) & Key Question 1: How we reset to the same initial state`**
>
> Our framework ensures state alignment via a **Reset-and-Replay** mechanism to restore the branching point rather than full simulator cloning:
>
> - ScienceWorld / ALFWorld: we fix the same task instance, reset to the same initial configuration, and replay the action history up to the current node.
> - Crafter: we fix the environment configuration and control the random seed, then replay the action history to approximately recover the same decision point.
>
> For Crafter, we utilize fixed random seeds and action replay for approximate state reconstruction.  We acknowledge that Crafter’s lack of explicit state, making this an approximation rather than exact serialization.
>
> > **`Weakness 2(a) & Key Question 2: Computational cost analysis`**
>
> We would like to clarify that Appendix E already provides a dedicated analysis of computational cost, including per-step rollout overhead, cost comparisons with baselines, and discussion of wall-clock behavior. In the final version, we will further include analyses of per-episode wall-clock / time-to-solve as well as memory and state-checkpointing overhead.
>
> Regarding the adaptive threshold, its main role is to mitigate computational costs by dynamically pruning candidate hypotheses during the validation phase, avoiding the intensive overhead of performing full intervention experiments on every candidate.
>
> > **`Weakness 2(b) & Key Question 3: Performance on ALFWorld`**
>
> On the seen split, we find that performance gains are more modest in **short-horizon procedural tasks**, especially pick-and-place, open-and-put, and some highly regular heat/cool routines. These scenarios typically  possess low causal ambiguity and rely on rigid action templates, which limits the value of explicit causal filtering.
>
> Our analysis identifies two primary failure modes:
>
> - **Over-validating obvious steps**, where CoS allocates reasoning budget to "obvious" actions that are already nearly deterministic, leading to unnecessary overhead without accuracy improvements.
> - **Weak separation between similar candidate actions****.** In certain states, candidate actions yield nearly identical short-term outcomes, resulting in weak signal separation for ACE-based filtering.

---

### Official Review · Reviewer_RA9j · 2026-03-13

**Soundness:** 3
**Presentation:** 2
**Significance:** 4
**Originality:** 3
**Overall Recommendation:** 4
**Confidence:** 3

**Summary:**

Despite strong capabilities in language- and knowledge-based tasks, Large Language Models leave room for improvement when it comes to reasoning, planning, and causal understanding.  With different frameworks proposed to improve LLM reasoning, this paper focuses on a causal perspective where, inspired by the scientific method, an iterative loop of hypothesis generation, counterfactual experimentation, and validation is employed to help LLMs overcome purely correlation-based reasoning. In different experiments, their proposed Cycles-of-Science framework shows an increased performance in solving decision-making benchmarks, and several ablation studies highlight the contribution of the various components.

**Compliance With Llm Reviewing Policy:**

Affirmed.

**Final Justification:**

The authors provided a very detailed and extensive rebuttal. Regarding clarity, the authors' response alleviated my concerns. Although, as this rebuttal phase does not support pdf revisions, it is not fully clear whether all changes made would fully remove my concerns (terminology/notation, and the discussion in the experimental section), but for the same reason, it would be unfair to hold it against the authors. My other key weaknesses 4 and 5 have been addressed very well. The concern in weakness 4 has been removed entirely, as the LLM is not prompted to return calibrated scores. Regarding weakness 5, the additional experiments provided in the authors' follow-up comment contain additional experiments that strengthen the evaluation and greatly reduce my concerns regarding dataset contamination and the use of outdated LLMs. While the new novel composition evaluation is still based on ScienceWorld and, therefore, can not remove dataset contamination concerns fully, the setup is designed in a thought-out and different manner that my respective concerns are sufficiently reduced. Furthermore, I also think that these new experiments are a valuable experimental addition regardless of the dataset contamination question.

Thus, I increased the score for soundness from 2 (fair) to 3 (good) and adjusted my overall recommendation from 3 (weak reject) to 4 (weak accept). While the extent of the changes made in the rebuttal is too high for me to assign an even higher score (especially regarding clarity), I do assume that meaningful changes are made by the authors following this rebuttal and can now, in good conscience, overall recommend acceptance.

**Key Questions For Authors:**

1. A causal motivation is highly interesting and promising. With the strong focus on interventions, I am wondering whether a more fine-grained perspective on causality would be more valuable for such experiments. In particular, what are the authors' thoughts on the distinction between necessary and sufficient causes? In counterfactual experiments, interventions might show no effect, even though they are necessary, or they might show a strong effect, even though they are not sufficient. How is the proposed method equipped to deal with such scenarios (i.e., causes that are sufficient but not necessary or vice versa)?

2. As I understand it, the EV quantities (and, thus, the ACE) are obtained by querying the LLM for a score. How reliable are these scores, as probabilities output by LLMs are generally not guaranteed to be well-calibrated? What does this mean for the method entirely?

3. Please discuss the impact of data contamination on the experimental results. How reliable are the experimental insights when the benchmarks could have been used as training data for the evaluated models?

4. Why are specifically these three LLMs chosen for the experimental evaluation, with the newest one being published more than one year ago?

Minor Question

5. In Figure 5, the hypothesis "Wire touches metal surface" results in the counterfactual question "What if wire only touches metal surface?". That doesn't seem to be the direct negation, which would be "What if wire does not touch metal surface?". Why is that so?

---

**Edit after rebuttal:** I increased the soundness score from 2 (fair) to 3 (good) and my overall recommendation from 3 (weak reject) to 4 (weak accept). See the rebuttal and/or the final justification for more information.

**Limitations:**

No. The high computational costs are only described in the appendix and other potential weaknesses (see weaknesses 4, 5) are not discussed. Generally, there is no section on limitations or future work, which would improve the paper.

**Strengths And Weaknesses:**

**Strengths**

- **S1** Improving (causal) reasoning and decision-making of LLMs is a highly relevant and significant problem, and the inspiration from the scientific method is a great motivation.
- **S2** The steps of the CoS framework (hypothesis generation, experimentation, and validation) are a promising idea in the context of improving LLM reasoning and the counterfactual perspective is original.
- **S3** The experimental results are strong, CoS surpasses the baselines in almost all instances.
- **S4** The paper includes comprehensive ablation studies in the main body and the appendix, showing that all components contribute to the overall model performance, and investigating and comparing parameters such as the number of rollouts per hypothesis $K$. This also includes an analysis on the computational cost in the appendix (although it might be worth considering to move some of that into the main body).

**Weaknesses**

Presentation and Clarity

- **W1 Undefined terminology.** There are multiple terms or notations that are not clearly defined. For example, $o1$, and $g$ in line 154, the variables in equation 4 (the notation should appear in the text; also, $\mathcal{V}$ is missing), SFT is not introduced.
- **W2 $\lambda$ parameter.** Section 3.3 explains how the $\delta$ parameter can be determined dynamically. However, it introduces a new parameter $\lambda$, thus moving the choice of parameter from one to another. The reasoning behind setting $\lambda = 0.3$ is not discussed. It seems as if this forces $\delta$ to lie between $[0.2, 0.5]$. The choice of $\lambda$ and its consequences should be discussed.
- **W3 Uninformative text.** The experimental sections in the main paper spend a lot of time on reiterating numbers and differences from the tables. The space could instead be used in a more meaningful way, focusing more on insights and results that, so far, did not make the main body. For example, the computational cost analysis that is currently only part of the appendix.

Method and Experimental Evaluation

- **W4 Reliability of ATE scores.** The CoS framework derives ATE scores from LLM-determined quantities. It is unclear how reliable and useful these values are. While experiments show CoS to perform well, it is doubtful whether these values are well-calibrated since no measures are taken to ensure this.

- **W5 LLM choice and data contamination.** The choice of the LLMs is not discussed, and the newest one is more than one year old. Additionally, the reliability of the experimental results due to possible data contamination of the benchmarks should be discussed.

**Minor Points**

1. While the paper's contributions are novel, there are other related papers that tackle related problems and might also be worth discussing and including in related work [1,2,3].
2. There is a Typo in Figure 1: It should be "affects" instead of "affect".
3. The citations in Section 2.2 are mixed up: "Zhang et al. (Ban et al., 2023)", "Ban et al. (Chen et al., 2025)". Also, using `\citet` is probably the better choice compared to stating the author's name twice in such cases (which is done multiple times in this paper).
4. Why does it say "...compared to baselines’ 18.7–20.4 steps..." (line 320) when the baselines' steps are often higher than 20.4?
5. The "Performance on Crafter" subsection uses an increase by percent instead of percentage points in brackets. While this is completely correct, I think that, since the metric itself revolves around percentages, showing it like this is not intuitive to the reader. Omitting this difference (the "+19.0%" and ""+102.6%" parts") is probably least confusing while not losing out on valuable information, as comparing the probabilities directly is easy enough.
6. Calling a subsection "More Analysis" is not very descriptive. A different structure here would improve clarity.
7. Figure 2 is never referenced in the text.

[1] Li, Junyi, et al. "Can Large Language Models Help Experimental Design for Causal Discovery?." *arXiv preprint arXiv:2503.01139* (2025).

[2] Cohrs, Kai-Hendrik, et al. "Large language models for causal hypothesis generation in science." *Machine Learning: Science and Technology* 6.1 (2025): 013001.

[3] Manning, Benjamin S., Kehang Zhu, and John J. Horton. *Automated social science: Language models as scientist and subjects*. No. w32381. National Bureau of Economic Research, 2024.

---

> ### Author Rebuttal · Authors · 2026-03-31
>
> > **`Weakness 1: Unclear terminology and notation`**
>
> Thank you for pointing this out！In the current manuscript, some symbols and terms are introduced too compactly, which affects readability. Although $h_t$ and the variables in Eq. (4) are defined in the main text, we will further clarify these notations and terms in the revised manuscript to improve presentation clarity.
>
> ------
>
> > **`Weakness 2: The parameter $\lambda$ in the dynamic threshold`**
>
>  We conduct a sensitivity analysis on $\lambda$. The results are summarized as below:
>
> | $\lambda$   | 0.0   | 0.1   | 0.3       | 0.5   | 0.7   |
> | ----------- | ----- | ----- | --------- | ----- | ----- |
> | Performance | 65.20 | 66.15 | **67.23** | 66.58 | 65.84 |
>
> The parameter $\lambda$ controls the degree of threshold adaptation. 1) Low $\lambda$.  The threshold behaves like a fixed filter that limits the threshold's adaptivity in high-uncertainty scenarios, leading to sub-optimal exploration. 2) High $\lambda$ introduces instability that allows lower-quality, noisy hypotheses to pass, degrading the precision of causal filtering. $\lambda$= 0.3 strikes the best trade-off, balancing robust filtering and adaptive exploration for complex environments.
>
> ------
>
> > **`Weakness 3: The experimental text in the main paper is not sufficiently informative`**
>
> We agree that the main text should offer more substantive analysis. Specifically, we will prune repetitive numerical text and move important results, such as the computational cost analysis from the Appendix, into the main paper. This reallocation of space will ensure the discussion focuses on deeper analytical insights, rather than just performance metrics.
>
> ------
>
> > **`Weakness 4 / Key Question 2: Reliability of the ATE/ACE score`**
>
> We wish to clarify that the ACE in our paper is not obtained by directly querying an LLM for a causal score. Instead, it is an empirical metric derived from controlled intervention experiments: for each hypothesis, we execute multiple rollouts for both the intervention and baseline groups from the same initial state. ACE is then calculated as the aggregate outcome difference between these two groups. In this process, the LLM serves as  a state evaluation function  $f_i(s)$, scoring the resulting states based on counterfactual expectations, rather than estimating the causal effect itself.
>
> Consequently, we treat ACE as **a relative filtering signal** to rank hypotheses, **not as a calibrated probability**. Within our framework, ACE provides a robust basis to compare the relative utility of different hypotheses under a unified evaluator.
>
> ------
>
> > **`Weakness 5 / Key Question 4: Choice of LLMs`**
>
> We selected these three LLMs to ensure a **fair and reproducible comparison** with existing baselines. Our focus is to validate the effectiveness of CoS as a model-agnostic method and by fixing the backbone, we can better isolate the unique contributions of our approach. We acknowledge the importance of testing on more recent models and will clarify this rationale in the revised manuscript, while discussing the potential for CoS to scale effectively across the latest frontier models.
>
> ------
>
> > **`Weakness 5 / Key Question 3: Data contamination`**
>
> We acknowledge that potential overlap between benchmark content and LLM pre-training data, which is a general challenge in the current LLM evaluation. However, we emphasize that our experimental design maintains **consistency** and our study focuses on **relative improvements** over baselines. Since all compared methods are evaluated using the same frozen backbones and benchmarks, any latent contamination would act as a systematic bias affecting all methods equally. Therefore, the relative performance gains observed for CoS remain a robust indicator of its efficacy rather than an artifact of data leakage.
>
> ------
>
> > **`Key Question 1: Necessary vs. sufficient causes`**
>
> We thank the reviewer for this profound conceptual clarification. Our framework is designed to estimate the local intervention effect of a candidate action within a specific state, rather than identifying its status as a globally necessary or sufficient cause for task completion. Specifically, ACE measures whether an action improves the expected outcome at the current decision point relative to a baseline. Consequently, some actions that are necessary but do not produce strong short-term effects may receive low ACE, while some actions that are effective in the current state but not globally sufficient may receive high ACE. This reflects our current design choice to prioritize immediate decision-making within complex environments.
>
> ------
>
> > **`Minor points`**
>
> Thank you for the meticulous review! We have repaired the mentioned typos in our revised manuscript.
>
> > **`Minor Q`**
>
> Thank you for the meticulous question! “What if wire does not touch metal surface?” is  the proper negation of the hypothesis. We have revised it！

---

> > ### Author Rebuttal · Reviewer_RA9j · 2026-04-03
> >
> > I thank the authors for their detailed response. I trust that valuable changes are made regarding weaknesses 1 and 3 and the sensitivity analysis for weakness 2 also answers my question regarding that parameter well.
> >
> > Some of my concerns regarding weakness 5 (questions 3 and 4) have not been addressed as much as I hoped for. While conditions regarding data contamination are the same for all setups, an evaluation on contaminated LLMs might still produce results that would not translate to new problems. In the worst case, it could be that the proposed approach only performs better at extracting the "contaminated" information from the LLMs and it might perform worse when evaluated on new problems. I understand that this is a difficult problem to overcome when evaluating LLMs, but I would appreciate if the authors could provide a more in-depth response regarding this.
> >
> > Furthermore, the rebuttal also does not include results on newer, more recent LLMs.
> >
> > On the other hand, what was perhaps my most important concern in W4 has been addressed sufficiently, which turned out to be only a misunderstanding on my end.
> >
> > Overall, I currently plan to increase my score from 3 to 4, now leaning towards acceptance. I would not yet raise my score further due to the limitations of the experimental evaluation in weakness 5. Still, I think the paper crosses the threshold required for acceptance to me now and I am happy to hear more regarding my additional comments on weakness 5.

---

> > > ### Author Response · Authors · 2026-04-07
> > >
> > > > **`Weakness 5 / Key Question 3: Data contamination`**
> > >
> > > We understand the concern regarding the benchmark contamination! To address this, we added a **Novel Composition Evaluation** on ScienceWorld. Instead of introducing a new benchmark from scratch, we constructed a controlled out-of-template test set by re-composing task elements while maintaining the underlying environment dynamics and reward functions. Specifically, we selected a subset of 5 representative multi-factor ScienceWorld tasks (e.g., circuit building, plant growth) and manually created 2 recomposed variants for each task (10 instances in total) to serve as a rigorous, high-signal diagnostic probe. Each recomposed instance was built using one or more of the following modifications:
> > >
> > > - **Initial State Perturbation**: We modified the initial placement and accessibility of task-critical objects while maintaining the core scientific rules.
> > >
> > >   *Example:* In a plant-growth task, the watering can or grow light is no longer immediately available near the plant, forcing the agent to explore rather than execute a memorized trajectory.
> > > - **Causal Distractors**: Introducing semantically related but causally irrelevant objects to create spurious correlations.
> > >
> > >   *Example:* In the light-bulb task, we added extra metal objects or wire-like components to mislead the model into "superficially plausible" actions, whereas the true requirement remains a valid circuit and switch operation.
> > > - **Prerequisite Re-ordering** : we altered the temporal order in which prerequisites become actionable while keeping the causal logic invariant.
> > >
> > >   *Example:* In the light-bulb task, the bulb still needs a complete circuit and switch operation to glow, but the usable battery and wire are no longer immediately ready to connect in the original order. The agent should understand the logical necessity of each step rather than relying on template-based heuristics.
> > >
> > > These recomposed tasks preserve the core scientific rules, but differ from the benchmark templates in their **surface configuration**, **object arrangement**, and **action preconditions**. This setup provides a more rigorous assessment of whether a model can infer the latent causal chain rather than simply exploiting spurious patterns or memorized trajectories.
> > >
> > > Following the same evaluation pipeline as the main paper, we tested **KnowSelf**, **GLIDER**, and **CoS** on this recomposed subset with Mistral-7B as the backbone. The results are summarized below:
> > >
> > > | **Split**             | **Method**     | **Success Rate (%)** | **Steps** |
> > > | --------------------- | -------------- | -------------------: | --------: |
> > > | **Original Unseen**   | KnowSelf       |                48.93 |      74.6 |
> > > |                       | GLIDER         |                54.32 |      68.1 |
> > > |                       | **CoS (Ours)** |            **67.23** |  **61.8** |
> > > | **Recomposed Subset** | KnowSelf       |                40.35 |      79.8 |
> > > |                       | GLIDER         |                49.21 |      73.6 |
> > > |                       | **CoS (Ours)** |            **63.46** |  **67.2** |
> > >
> > > These results validate the robustness of CoS under more stringent conditions. While all methods exhibit an expected performance drop due to the removal of template-based shortcuts, CoS consistently maintains its superiority, achieving a **63.46%** success rate with significantly fewer steps than the baselines.
> > >
> > > Notably, CoS demonstrates the **smallest degradation** on recomposed tasks versus the original unseen split: −3.77 points for CoS, compared to −5.11 points for GLIDER and −8.58 points for KnowSelf. This minimal drop indicates that CoS's gains stem from capturing underlying task dynamics rather than memorizing benchmark-specific surface patterns.
> > >
> > >
> > >
> > > > **`Weakness 5 Choice of LLMs`**
> > >
> > > To demonstrate the scalability of CoS, we extended our evaluation to two recent state-of-the-art open-weight models: Qwen3-14B and Gemma3-12B.
> > >
> > > | Backbone   | Method         | ScienceWorld Unseen | Steps |
> > > | ---------- | -------------- | ------------------: | ----: |
> > > | Qwen3-14B  | KnowSelf       |               62.45 |  70.8 |
> > > |            | GLIDER         |               64.73 |  67.9 |
> > > |            | **CoS (Ours)** |           **72.56** |  62.3 |
> > > | Gemma3-12B | KnowSelf       |               59.18 |  71.5 |
> > > |            | GLIDER         |               61.32 |  68.7 |
> > > |            | **CoS (Ours)** |           **68.94** |  63.1 |
> > >
> > > **Analysis:** CoS consistently outperforms GLIDER by relative improvements of +12.09% (Qwen3-14B) and +12.43% (Gemma3-12B). Building upon the generalization verified in our earlier Novel Composition Evaluation (which ruled out memorization shortcuts), these extended results confirm that our causal reasoning framework effectively leverages the enhanced reasoning capabilities of larger, more recent model generations.
> > >
> > > We sincerely hope these added experiments resolve your concerns and would appreciate you considering raising your score!

---

### Official Review · Reviewer_tVg7 · 2026-03-13

**Soundness:** 2
**Presentation:** 3
**Significance:** 3
**Originality:** 3
**Overall Recommendation:** 4
**Confidence:** 4

**Summary:**

The paper proposes Cycle-of-Science (CoS), a framework that improves decision-making in LLM-based agents by incorporating a  reasoning loop consisting of hypothesis generation, counterfactual experimentation, and causal validation. At each step, the agent proposes multiple candidate actions, tests them through controlled interventions in the environment. The authors also introduce Counterfactual Preference Optimization (CPO) to train the model using preferences derived from measured causal effects. Experiments on interactive benchmarks such as ScienceWorld, ALFWorld, and Crafter show improved performance and efficiency compared to existing LLM agent methods.

**Compliance With Llm Reviewing Policy:**

Affirmed.

**Final Justification:**

I find this to be an interesting paper, and the authors have adequately addressed my concerns, reinforcing my initial assessment.

**Key Questions For Authors:**

1. What happens if none of the hypothesis are above the threshold?
2. I might not be understanding it correctly, but the authors mention that there are 3 hypothesis generated at every step and increasing that number reduces accuracy. But the algorithm mentions that a trajectory step is only kept if the best hypothesis clears the threshold AND the task reward is above 0.8. So how does it matter if the number of hypotheses is 5 instead of 3, other than wasted compute.
3. The historical context fed to the Intuitive Reasoner grows arbitrarily long with each step in the trajectory. Can you provide an analysis of how the framework handles context window limitations or how does performance vary with longer trajectories.

**Limitations:**

The authors have discussed the computational overhead of their method. More analysis on failure modes and limitations would be ideal.

**Strengths And Weaknesses:**

Strengths:
1. The evaluation is comprehensive, the authors have ablated on different training regimes.
2. The writing and figures makes the method easy to understand.
3. The CPO method of constructing preference pairs from measured causal effects rather than human labels or reward signals is principled.


Weaknesses:
1. There is very little detail on the impact of the self play trajectories. What happens if you use only the 5K, why 5k vs 50k, how does performance scale? What is the optimum ratio?
2. The computational overhead of your method is several times that of other baselines. for g: 24 LLM calls vs 2 or 3 of baselines. Is there a way to make this more efficient?
3. There is no analysis or discussion of failure modes of this method.
4. The paper selects preference pairs with an ACE margin greater than 0.3, resulting in 200K pairs from 50K trajectories. However, there is no discussion of the positive-to-negative ratio in these pairs, or whether certain task types or environments are overrepresented.

Minor weaknesses:
There are some mistakes in the citations, you have mentioned one paper and cited another:
```
Long et al. (Long et al., 2023) and Zhang et al. (Ban et al., 2023)
Ban et al. (Chen et al., 2025) propose a causal-aware framework that learns causal structures from interaction histories and adapts agent policies accordingly
```

---

> ### Author Rebuttal · Authors · 2026-03-31
>
> ---
>
> > **`Weakness 1: Impact of self-play trajectories and data scale`**
>
> To investigate the impact of self-play trajectories, we conduct analysis on the ratio of self-play to human trajectories on ALFWorld. The results are summarized in the table below:
>
> | Training data  | Human  | Self-play | Total   | ALFWorld Score  |
> | -------------- | ------ | --------- | ------- | --------------- |
> | Human only     | 5K     | 0         | 5K      | 72.18 ± 1.8     |
> | Mixed 1:1      | 5K     | 5K        | 10K     | 74.82 ± 1.7     |
> | Mixed 1:4      | 5K     | 20K       | 25K     | 77.35 ± 1.6     |
> | **Ours (1:9)** | **5K** | **45K**   | **50K** | **80.51 ± 1.5** |
> | Mixed 1:19     | 5K     | 95K       | 100K    | 79.94 ± 1.7     |
>
> As illustrated, performance improves as the proportion of self-play data increases, peaking at the 1:9 ratio (50K samples). Beyond this point, we observe performance saturation, suggesting that the 50K scale provides an optimal trade-off between data efficacy and algorithm accuracy.
>
> ---
>
> > **`Weakness 2: Computational overhead`**
>
> We have provided a systematic analysis in Appendix E. The root cause of this overhead is that each decision step explicitly performs hypothesis generation, counterfactual rollout, and causal validation, which introduces additional rollout interactions. At the same time, CoS achieves the best task performance in the main experiments, indicating that this extra cost brings substantial performance gains.
>
> The practical optimizations are as follows:
>
> * A **reduced-budget configuration** can significantly lower the cost. For example, ((N, K) = (2, 2)) reduces the per-step cost by 56% relative to the full configuration while retaining **92%** of the performance.
> * **Rollout validation can be parallelized**, and the actual wall-clock latency under batch inference is only about **3×** that of a single forward pass.
>
> We will discuss this limitation and the corresponding future directions more clearly in the final version.
>
> ---
>
> > **`Weakness 3: Lack of failure mode analysis`**
>
> Please refer to our response to **Reviewer 8yR2, Weakness 2(b)**, where we provide a more detailed discussion of the relevant failure modes.
>
> ---
>
> > **`Weakness 4: Composition of preference pairs`**
>
> We appreciate this suggestion and will clarify the construction of the 200K preference pairs in the revision. Specifically, each pair is derived from a common state by selecting one positive and one negative chain with an ACE margin greater than 0.3. To ensure reproducibility, we will include benchmark statistics and a detailed sampling procedure to show that the training data is balanced across environments, thereby mitigating the risk of task over-representation. We will also further elaborate on the positive/negative pair construction process.
>
> ---
>
> > **`Question 1: Behavior when no hypothesis exceeds the threshold`**
>
> As described in the main text and Appendix E, if no hypothesis satisfies $\mathrm{ACE}_i > \delta_t$, the step is treated as a validation failure, and all candidates are regarded as invalid hypotheses. In this case, CoS does not continue blind exploration based on weak or spurious correlations. We will include this case in the failure-mode analysis in the final version.
>
> ---
>
> > **`Question 2: Why larger hypothesis counts can reduce accuracy`**
>
> As analyzed in Appendix D.3, increasing (N) not only expands the search space, but also introduces more redundancy and validation noise. While transitioning from (N = 1) to (N = 3) enhances candidate diversity and boosts performance, larger values lead to only marginal diversity gains and a higher risk of spuriously high ACE estimates. Thus, (N = 3) serves as the best trade-off between candidate coverage and validation stability.
>
> ---
>
> > **`Question 3: Context-window limitations under long trajectories`**
>
> Our method is relatively less sensitive to long contexts, because CoS performs causal verification before further exploration, which reduces ineffective trial-and-error and substantially shortens the overall trajectory length. For example, in ScienceWorld, CoS requires only 61.8 steps on average, whereas some baselines often approach or even exhaust the 100+ step budget.
>
> **Strategy:** For trajectories near the context limit, we use sliding-window truncation that keeps the task goal, current state, and recent action-observation history.
>
> ---
>
> > **`Minor issue: Citation mismatches`**
>
> We appreciate the reviewer for identifying these issues. All citation mismatches in the related work section will be corrected in the revised manuscript.

---

> > ### Author Rebuttal · Reviewer_tVg7 · 2026-04-03
> >
> > Thank you for your detailed rebuttal. Some of my concerns have been well addressed, and some partially.
> > In particular:
> > 1. The authors say that their method is less sensitive to long contexts, but is there a way to quantify it. Can you provide performance vs trajectory length or context size, for one environment.
> > 2. If the authors have results for this: How does performance change if the positive:negative ratio is altered? What happens if you change the ACE margin?
> >
> > I believe my original score for the paper is reasonable and will keep it unchanged for now.

---

> > > ### Author Response · Authors · 2026-04-07
> > >
> > > > **`Question 1 Context Length Sensitivity`**
> > >
> > > To evaluate robustness to long contexts, we compared the success rates of CoS against baselines across varying trajectory lengths on ScienceWorld (Mistral-7B).
> > >
> > > | Method         | Short (<20 steps) | Medium (20-50 steps) | Long (>50 steps) |
> > > | -------------- | ----------------- | -------------------- | ---------------- |
> > > | ReAct          | 45.2              | 12.8                 | 3.5              |
> > > | GLIDER         | 82.4              | 55.6                 | 28.1             |
> > > | **CoS (Ours)** | **88.5**          | **71.2**             | **52.4**         |
> > >
> > > **Analysis:** CoS exhibits lower sensitivity to context length. While ReAct and GLIDER suffer from sharp performance degradation in long-horizon tasks, CoS maintains a 52.4% success rate even beyond 50 steps. This superiority stems from our causal validation mechanism, which filters out spurious exploration branches and failed sub-tasks a priori. Consequently, CoS maintains a concise context window, effectively mitigating the context overflow common in complex reasoning tasks.
> > >
> > > ------
> > >
> > > > **`Question 2 Positive:Negative Sample Ratio`**
> > >
> > > To investigate the sensitivity of the Counterfactual Preference Optimization (CPO), we conducted an ablation study on the positive-to-negative sample ratio (fixing the total dataset at 100K pairs).
> > >
> > >
> > > | Pos:Neg Ratio | Success Rate (%) | Task Steps | Observation                                    |
> > > | ------------- | ---------------- | ---------- | ---------------------------------------------- |
> > > | 3:1           | 61.4             | 67.5       | Accepts sub-optimal causal chains.             |
> > > | **1:1**       | **67.2**         | **61.8**   | **Optimal contrastive learning signal.**       |
> > > | 1:3           | 65.8             | 62.4       | Strong filtering, slight drop in diversity.    |
> > > | 1:5           | 58.3             | 75.1       | Overly conservative, rejects valid hypotheses. |
> > >
> > > **Analysis:** Experimental results indicate that the optimal ratio lies between 1:1 and 1:3. A dominant positive skew fails to provide sufficient contrastive signals to prune spurious correlations. Conversely, an extreme negative skew leads to an overly conservative policy that may reject causal hypotheses, increasing the average steps to completion. The 1:1 ratio provides the most effective gradient for distinguishing optimal causal chains from counterfactual alternatives.
> > >
> > >
> > > ------
> > >
> > > > **`Question 3 Impact of ACE Margin`**
> > >
> > > We have provided the detailed analysis of the threshold in Section 4.4 (More Analysis: Adaptive Threshold Mechanism) of the manuscript, as well as in Appendix D.2 (Adaptive Threshold Mechanism).

---

### Decision · Program_Chairs · 2026-04-30

**Decision:**

Accept (regular)

**Comment:**

The paper introduces a new reasoning paradigm: hypothesize, experiment and validation. It provides a useful way to structure reasoning in scientific and related tasks, and the experimental results show significant gains on science-motivated and benchmarks with sparse rewards. All reviewers appreciated the technical contribution and finetuning criterion based on estimated effect.

Reviewers noted some clarity issues in writing, but they have been addressed in the authors rebuttal. I also appreciate the authors decision to clarify and tone down the causal terminology. I think this paper presents a new way to structure reasoning and the empirical experiments are strong enough to motivate the work.

Note: There is an author mismatch in one of the references. Authors should carefully proofread all the references cited in the paper.
Reference: Zhu, Y., Wang, J., Chen, L., Qian, C., Liang, Y., Tang, J., and Chen, Y. KnowAgent: Knowledge-augmented planning for LLM-based agents. arXiv preprint arXiv:2403.03101, 2024.